# TRIB3 supports breast cancer stemness by suppressing FOXO1 degradation and enhancing SOX2 transcription

Jin-mei Yu [1,4], Wei Sun[1,4], Zhen-he Wang[1,4], Xiao Liang[1], Fang Hua [1], Ke Li[2], Xiao-xi Lv[1], Xiao-wei Zhang [1], Yu-ying Liu[3], Jiao-jiao Yu[1], Shan-shan Liu[1], Shuang Shang[1], Feng Wang[1], Zhao-na Yang[1], Chen-xi Zhao[1], Xue-ying Hou[1], Ping-ping Li[1], Bo Huang[3], Bing Cui [1]* & Zhuo-Wei Hu[1]*

The existence of breast cancer stem cells (BCSCs) is a major reason underlying cancer metastasis and recurrence after chemotherapy and radiotherapy. Targeting BCSCs may ameliorate breast cancer relapse and therapy resistance. Here we report that expression of the pseudokinase Tribble 3 (TRIB3) positively associates with breast cancer stemness and progression. Elevated TRIB3 expression supports BCSCs by interacting with AKT to interfere with the FOXO1-AKT interaction and suppress FOXO1 phosphorylation, ubiquitination, and degradation by E3 ligases SKP2 and NEDD4L. The accumulated FOXO1 promotes transcriptional expression of SOX2, a transcriptional factor for cancer stemness, which in turn, activates FOXO1 transcription and forms a positive regulatory loop. Disturbing the TRIB3-AKT interaction suppresses BCSCs by accelerating FOXO1 degradation and reducing SOX2 expression in mouse models of breast cancer. Our study provides insights into breast cancer development and confers a potential therapeutic strategy against TRIB3-overexpressed breast cancer.

[1] State Key Laboratory of Bioactive Substance and Function of Natural Medicines, Institute of Materia Medica, Chinese Academy of Medical Sciences and Peking Union Medical College, Beijing 100050, China. [2] Institute of Medicinal Biotechnology, Chinese Academy of Medical Sciences and Peking Union Medical College, Beijing 100050, China. [3] Institute of Basic Medical Sciences, Chinese Academy of Medical Sciences and Peking Union Medical College, Beijing 100005, China. [4] These authors contributed equally: Jin-mei Yu, Wei Sun, Zhen-he Wang. *email: cuibing@imm.ac.cn; huzhuowei@imm.ac.cn

Breast cancer is composed of heterogeneous cell populations that interact in complex networks[1]. Cancer stem cells (CSCs) play key roles in intra- and intertumoral heterogeneities, which are responsible for tumor progression, resistance to therapy, and disease relapse[2]. Breast CSCs (BCSCs) represent a dynamic subpopulation of breast cancer cells (BCCs), which have the capabilities of self-renewal, tumor initiation, and the ability to give rise to more differentiated progeny upon xenotransplantation into immunocompromised mice[3,4]. CSC markers, including pluripotency genes with normal stem cells in the tissue-of-origin, often share transcriptional profiles[4]. Recent work has indicated the plasticity of BCSCs, revealing that these cells exist in two distinct but interchangeable states: epithelial–mesenchymal transition–CSCs, a quiescent mesenchymal type marked as $EpCAM^-CD49f^+CD44^+CD24^-$, and mesenchymal–epithelial transition–CSCs, an epithelial, proliferative type identified as $EpCAM^+CD49f^+ALDH^+$ [2,5]. Plastic CSCs are not necessarily rare and/or quiescent, and niche signals may instruct them following neutral competition dynamics. Differentiated cells and transient-amplifying cells can be reprogrammed into CSCs in the niche via plastic mechanisms[6]. Evidence suggests that CSCs are always recreated as long as the CSC niche remains intact. Hence, modulating CSC niche functions has become an attractive alternative rather than directly targeting CSCs[6].

The pseudokinases Tribbles homolog 3 (TRIB3, NIPK, SIKP3) is one of three mammalian homologs of *tribbles* in *Drosophila*, which inhibit mitosis in embryo and germ cell development[7]. TRIBs are fundamental regulators of cellular stress, the cell cycle, differentiation, and proliferation[8]. Cisplatin-enriching lung CSCs show activated TRIB1/HDAC activity[9]. Mice reconstituted with Trib2 by the engraftment of hematopoietic stem cells retrovirally expressing Trib2 uniformly developed fatal transplantable acute myelogenous leukemia[10,11]. TRIB3 has recently been identified as a stress sensor in response to various tumor microenvironments or niche-rich stressors, including amino acid or glucose deficiency, insulin, unfolded protein accumulation in the endoplasmic reticulum, and oxidative damage[8,12,13]. TRIB3 promotes chronic inflammation and cancer by interacting with intracellular signaling and functional proteins. The interaction of TRIB3 and the autophagic receptor p62 interferes with the degradation of autophagy and the ubiquitin-proteasome system to control the initiation and progression of cancer[14]. TRIB3 promotes PML-RARα-driven acute promyelocytic leukemia by interacting with PML-RARα and regulating PML-RARα degradation[15]. These findings are consistent with the findings that enhanced TRIB3 expression negatively associates with overall survival in colorectal cancer[16], breast cancer[17,18], and gastric cancer[19]. Given that TRIB3 senses a variety of stress signals, and that enhanced TRIB3 expression leads to poor prognosis for breast cancer patients, we postulated that TRIB3 contributes to the pathogenesis of breast cancer via its tumor initiation capacity. We studied the coordinative functions and mechanisms of TRIB3, Sry-related high-mobility box 2 (SOX2), and Forkhead box O1 (FOXO1) in supporting BCSCs, and elucidated the implications of these findings.

## RESULTS

**TRIB3 is associated with the stemness of breast cancer**. We queried the Curtis breast dataset from the Oncomine database of BCCs for information on 1556 patients with invasive ductal breast cancer and 148 patients with invasive lobular breast cancer. These patients expressed higher levels of *TRIB3* than their normal counterparts ($n = 144$) (Fig. 1a). Using human tumor tissue microarrays of breast adenocarcinoma, higher TRIB3 levels were observed in tumor tissues than in adjacent nontumor tissues

(Fig. 1b). TRIB3 expression showed no stage differences among a variety of breast cancer patients. We then re-interrogated the *TRIB3* expression in the published GSE12790 dataset and no expression differences were found among the luminal, Her2-amplified, and basal BCC lines (Supplementary Fig. 1a). TRIB3 is universally highly expressed in HMLER and eight distinct breast cancer epithelial cell lines but not in human mammary epithelial cells (HMLEs) (Fig. 1c). We interrogated The Cancer Genome Atlas (TCGA) database using online kmplot tools, to evaluate BCCs from 1117 patients with breast adenocarcinoma[20]. The patients were divided into three groups based on their relative *TRIB3* expression levels in BCCs. Patients with tumors expressing *TRIB3* mRNA in the upper tertile ($TRIB3_H$) had a significantly shorter overall survival than patients with tumors expressing *TRIB3* mRNA at the lower and intermediate tertile levels ($TRIB3_{L+M}$) ($p = 0.0002$; Fig. 1d). We then queried the PubMed GEO database to evaluate BCCs from 582 patients with breast adenocarcinoma as previously described[21]. Approximately two of three of these patients (426 of 582) did not have detectable cancer in the regional lymph nodes at the time of surgery and were not administered adjuvant therapy. The remaining patients had the detectable disease in regional lymph nodes and received adjuvant therapy. Among 582 patients, 46% relapsed ($n = 270$) and had a median metastasis-free survival time of 22.1 months. The relative level of *TRIB3* in BCCs was used to segregate patients into three groups (Supplementary Table 1). Patients with tumors expressing *TRIB3* mRNA at the upper tertile level ($TRIB3_H$) had a significantly shorter metastasis-free survival time than patients with tumors expressing *TRIB3* at the lower ($TRIB3_L$) or intermediate tertile ($TRIB3_M$) levels ($p < 0.0001$; Fig. 1e). Moreover, patients with $TRIB3_H$ tumors had lower rates of overall survival (TCGA-BRCA; Supplementary Fig. 1b) and metastasis-free survival (PubMed GEO Datasets; Supplementary Fig. 1c) among ER+, HER2+, and triple-negative breast cancer patients. Using a multivariate Cox regression model, high *TRIB3* expression was found to have a predictive value for short metastasis-free survival (Supplementary Table 2). These data indicate that elevated TRIB3 expression positively correlates with breast cancer progression, metastasis, and relapse.

Inflammation, hypoxia, and metabolic stresses play vital roles in the progression of breast cancer by affecting key CSC phenotypes[1,2,22,23]. We found that cytokines interleukin (IL)-6, transforming growth factor-β, IL-1β, tumor necrosis factor-α, and hypoxia, as well high glucose enhanced TRIB3 expression in MCF7 cells (Supplementary Fig. 1d), suggesting that TRIB3 acts as a stress sensor in response to a diverse range of stressors. The cytokine IL-6 was chosen for the following experiments due to its potent effects on TRIB3 induction. IL-6 treatment increased the tumorsphere-formation ability of MCF7 cells (Supplementary Fig. 1e). IL-6 enhanced TRIB3 expression in MCF7 and MDA-MB-231 cells in dose- (Supplementary Fig. 1f) and time-(Supplementary Fig. 1g) dependent manners. Silencing TRIB3 expression reduced mammosphere formation in MCF7 cells with or without IL-6 stimulation (Supplementary Fig. 1h). In isolated mammary epithelial cells (MECs) from spontaneous breast cancer mice, high TRIB3 or *Trib3* expression was found in the $CD24^+$ $CD29^{low}$ subpopulation (Fig. 1f, g) and $CD24^+CD90^+$ stem-like subpopulation (Supplementary Fig. 1i) from mice with luminal-type MMTV-PyMT breast cancer, in the $CD24^+CD29^{high}$ subpopulation from mice with her2-type MMTV-ErbB2 breast cancer (Fig. 1h, i), and in the $CD68^+$ tumor-associated macrophage (TAM) adjacent area in breast cancer patients (Supplementary Fig. 1j). In TAMs and MECs co-culture assays (Supplementary Fig. 1k left), silencing *Trib3* in MECs had no effect on IL-6 production from MMTV-PyMT-derived TAMs

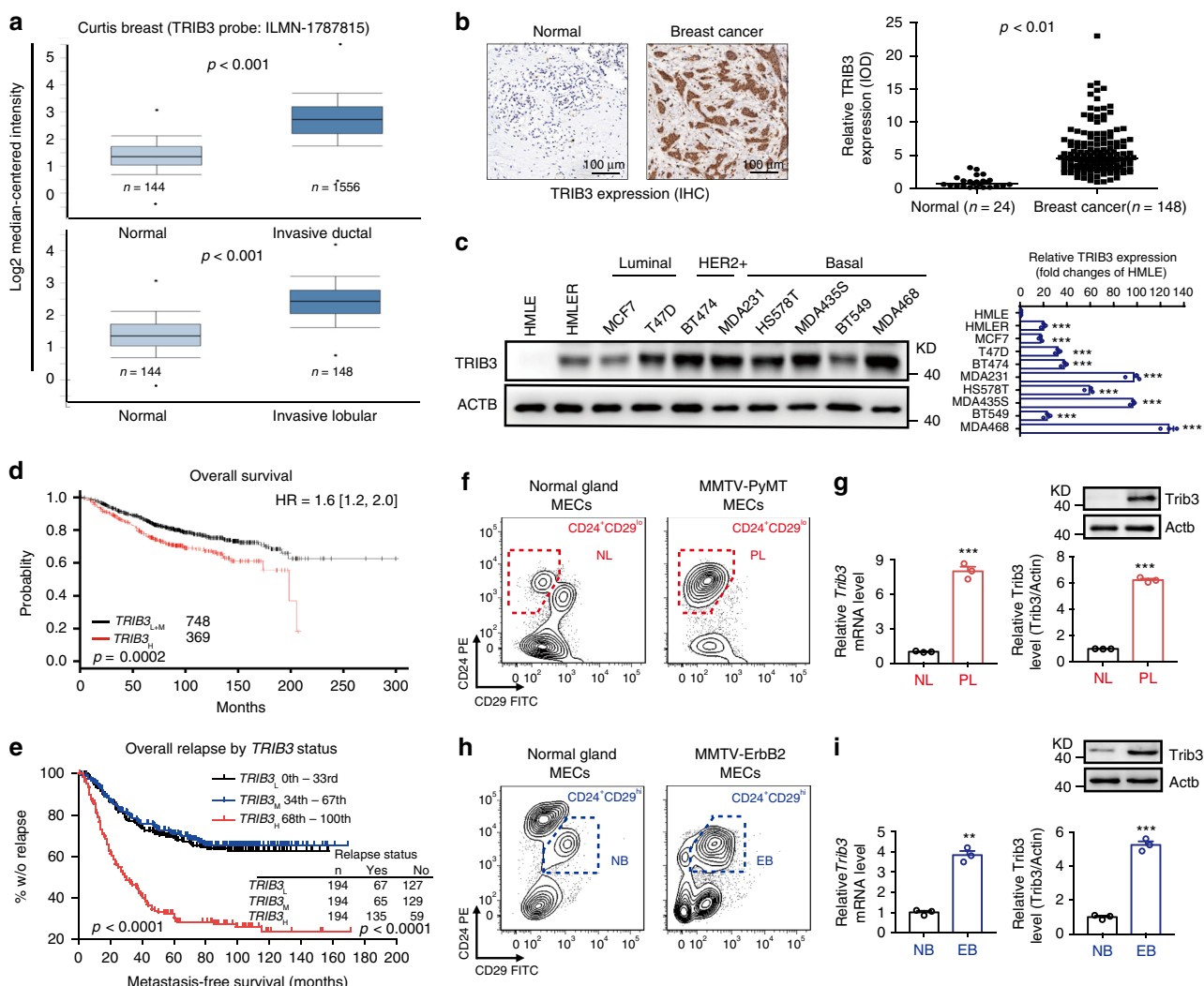

**Fig. 1 High TRIB3 expression is negatively associated with overall survival and metastasis-free survival in breast cancer. a** Graph derived from published data available in the Oncomine database. The box charts depict the relative expression of *TRIB3* in invasive ductal (top) and invasive lobular (bottom) breast cancer patients. Centre line = 50th percentiles; bounds of box = 25th and 75th percentiles; bars = 10th and 90th percentiles; whiskers = min and max values. **b** Formalin-fixed, paraffin-embedded tissue microarray sections of normal and breast cancer tissues were stained with an anti-TRIB3 antibody. Tissue-bound TRIB3 is shown in brown (left). The dot plots show the relative expression of TRIB3 calculated by the intensity of the brown color in tissue array immunohistological images (right). **c** Immunoblots of protein lysates from HMLE, HMLER, and BCCs, as indicated at the top of the left panel. The relative TRIB3 expression quantification from three independent studies is shown (right). **d, e** Graph derived from TCGA (**d**) or published data (**e**) available in the PubMed GEO database (GSE2603, GSE5327, GSE2034, and GSE12276). For each analysis, 1117 patients (**d**) or 582 patients (**e**) were segregated into tertiles, with the group designated *TRIB3*_H representing one-third of the patients who had tumors with the highest levels of *TRIB3* mRNA and the group designated *TRIB3*_L representing one-third of patients who had cancers with the lowest levels of *TRIB3* mRNA. One-third of patients who had tumors with intermediate *TRIB3* mRNA expression were designated as *TRIB3*_M. Overall survival (**d**) and metastasis-free survival (**e**) were determined by Kaplan–Meier analyses, and significant differences were determined by the log-rank test. The number of patients in each category, the total metastatic events, and the corresponding P-values ($\chi^2$-test) are shown in the embedded tables. **f-i** Flow cytometry of CD45⁻CD31⁻TER119⁻(Lin⁻) MECs from the mammary glands of 6-week-old MMTV-PyMT female mice (**f**) and 6-month-old MMTV-ErbB2 female mice (**h**). Relative *Trib3* mRNA expression and TRIB3 protein expression are indicated in **g** and **i**. Data are presented as the mean ± SEM; $P > 0.05$ was considered not significant (NS); *$P < 0.05$, **$P < 0.01$, and ***$P < 0.001$ compared with the HMLE group in **c** and the normal gland group in **g** and **i**. Source data are provided as a Source Data file.

(Supplementary Fig. 1k middle), but reduced mammosphere formation (Supplementary Fig. 1k right). These data indicate that elevated TRIB3 expression links BCSC-promoting cytokines and other stressors to breast cancer stemness.

We ranked 20,530 genes from breast cancer samples in the TCGA dataset by their relative *TRIB3* expression in the top 10th percentile (*TRIB3*^Hi) vs. the bottom 10th percentile (*TRIB3*^Low) for gene-set enrichment analyses. *TRIB3*^Hi tumor samples were enriched in the expression of gene signatures associated with "ES/Stem cells" in comparison with

*TRIB3*^Low samples (Fig. 2a). Three-dimensionally (3D) cultured CD44^bright CD24^dim (CD44^Br CD24^Di) and non-CD44^bright CD24^dim (non-CD44^Br CD24^Di) MCF7 cells (Fig. 2b) were isolated, and each subset was examined for TRIB3 mRNA and protein expression, and stemness-related proteins, including SOX2, NANOG, Octamer-binding transcription factor 4 (OCT4), Kruppel-like factor 4 (KLF4), and c-Myc. The mRNA expression of *TRIB3*, *SOX2*, *NANOG*, *OCT4*, *KLF4*, and *c-Myc* was higher in CD44^Br CD24^Di cells than in non-CD24^Di CD44^Br cells (Fig. 2c). However, higher protein expression of TRIB3,

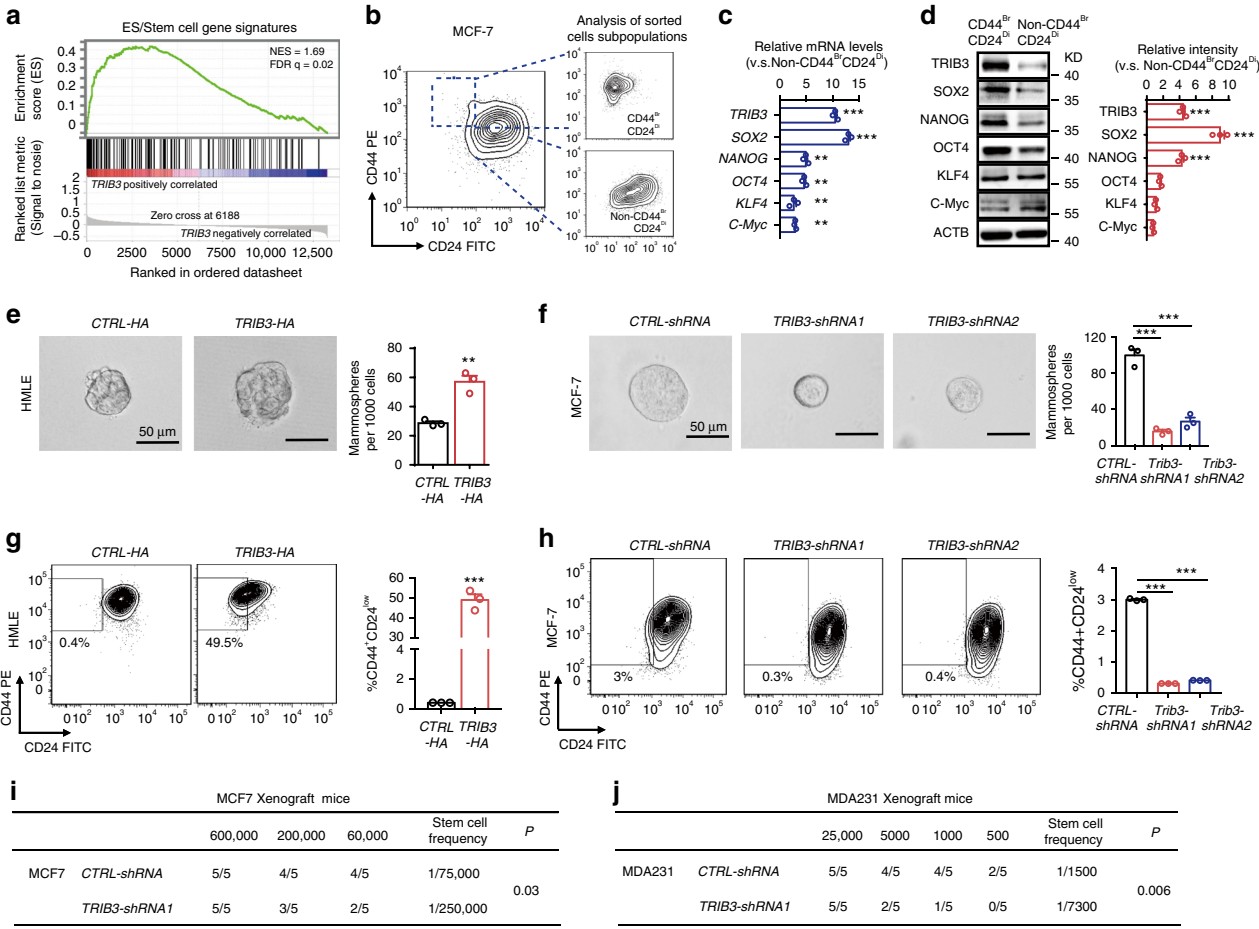

**Fig. 2 Silencing TRIB3 inhibits the spheroid formation and tumor engraftment. a** GSEA demonstrating the enrichment of gene sets related to ES stem cell gene signatures in the ranked gene list of the top 10th percentile ($n = 121$) vs. the bottom 10th percentile ($n = 121$) of breast cancer patients expressing *TRIB3* from the TCGA database ($n = 1215$). NES, normalized enrichment score. FDR, false discovery rate. **b** Flow cytometric analyses of representative unsorted MCF7 cells (left), isolated CD44$^{Br}$CD24$^{Di}$ MCF7 cells (upper right), and isolated non-CD44$^{Br}$CD24$^{Di}$ MCF7 cells (lower right). Contour plots (10% probability) depict the fluorescence of BCCs stained with fluorochrome-conjugated mAbs specific for human CD44 (y axis) or human CD24 (x axis). **c**, **d** Gene expression (**c**) by real-time PCR or immunoblots (**d**) of protein lysates of CD44$^{Br}$CD24$^{Di}$ and non-CD44$^{Br}$CD24$^{Di}$ MCF7 cells. The height of each column in the histogram indicates the fold increase in each gene or protein, as indicated on the left, of sorted CD44$^{Br}$CD24$^{Di}$ MCF7 cells relative to that of sorted non-CD44$^{Br}$CD24$^{Di}$ MCF7 cells. **e**, **f** Tumor spheroid formation of HMLE cells (**e**) transfected with control vector or TRIB3-expressing vector (as indicated at the top) or MCF7 cells (**f**) transfected with either *CTRL-shRNA* or *TRIB3-shRNA*. All data are shown as the number of mammospheres per 1000 cells. **g**, **h** Flow cytometric assays detecting the percentage of CD44$^{Br}$CD24$^{Di}$ in HMLE cells transfected with a control vector or a TRIB3-expressing vector (**g**), or MCF7 cells transfected with either *CTRL-shRNA* or *TRIB3-shRNA* (**h**). **i**, **j** Tumor incidence in animals implanted with MCF7 cells (**i**) or MDA-MB-231 cells (**j**) transfected with either *CTRL-shRNA* or *TRIB3-shRNA*. Data are shown as the mean ± SEM; $P > 0.05$ was considered not significant (NS); \*$P < 0.05$, \*\*$P < 0.01$, and \*\*\*$P < 0.001$, compared with the vector or *CTRL-shRNA* group. Source data are provided as a Source Data file.

SOX2, and NANOG was found in CD44$^{Br}$CD24$^{Di}$ cells than that in non-CD24$^{Di}$CD44$^{Br}$ cells (Fig. 2d). Moreover, *Trib3*-silenced MECs from MMTV-PyMT transgenic mice and MMTV-ErbB2 transgenic mice formed much fewer spheroids than the control MECs (Supplementary Fig. 2a, b). Transfection of TRIB3-negative HMLE cells with TRIB3 enhanced spheroid formation (Fig. 2e) and the CD44$^{Br}$CD24$^{Di}$ subpopulation (Fig. 2g). The TRIB3-silenced cells exhibited a reduced capacity for spheroid formation (Fig. 2f and Supplementary Fig. 2e) and a reduced number of cells in the stem-like CD44$^{Br}$CD24$^{Di}$ subpopulation (Fig. 2h). In addition, silencing TRIB3 expression suppressed tumor growth (Supplementary Fig. 2d, f) and metastasis (Supplementary Fig. 2g). Finally, TRIB3-silenced BCCs reduced the capacity to form tumors in immunodeficient mice (Fig. 2i, j). These data suggest that elevated TRIB3 expression positively correlates with the spheroid formation in vitro and tumor engraftment efficiency in vivo.

**TRIB3 enhances SOX2 expression via FOXO1 transcription.** We examined the expression of stemness-associated proteins in TRIB3-negative HMLE cells overexpressing TRIB3 (Fig. 3a top and Fig. 3b) and TRIB3-positive MCF7 (Fig. 3a bottom and Fig. 3c) and MDA-MB-231 (Supplementary Fig. 3a, b) cells in which TRIB3 expression was silenced. Overexpression of TRIB3 in HMLE cells dramatically enhanced the mRNA expression of *SOX2* (Fig. 3a top) and promoted protein expressions of SOX2, NANOG, and c-Myc (Fig. 3b). Silencing TRIB3 expression in MCF7 cells attenuated the mRNA expressions of *SOX2* and *POU5F1* (Fig. 3a bottom) and protein expressions of SOX2 and NANOG (Fig. 3c); in MDA-MB-231 cells, silencing TRIB3 expression intensively attenuated the mRNA expression of *SOX2* and *KLF4* (Supplementary Fig. 3a), and inhibited protein expression of SOX2 and c-Myc (Supplementary Fig. 3b). SOX2 was universally correlated with TRIB3 expression in all three types of cells. Moreover, silencing *SOX2* but not other pluripotency

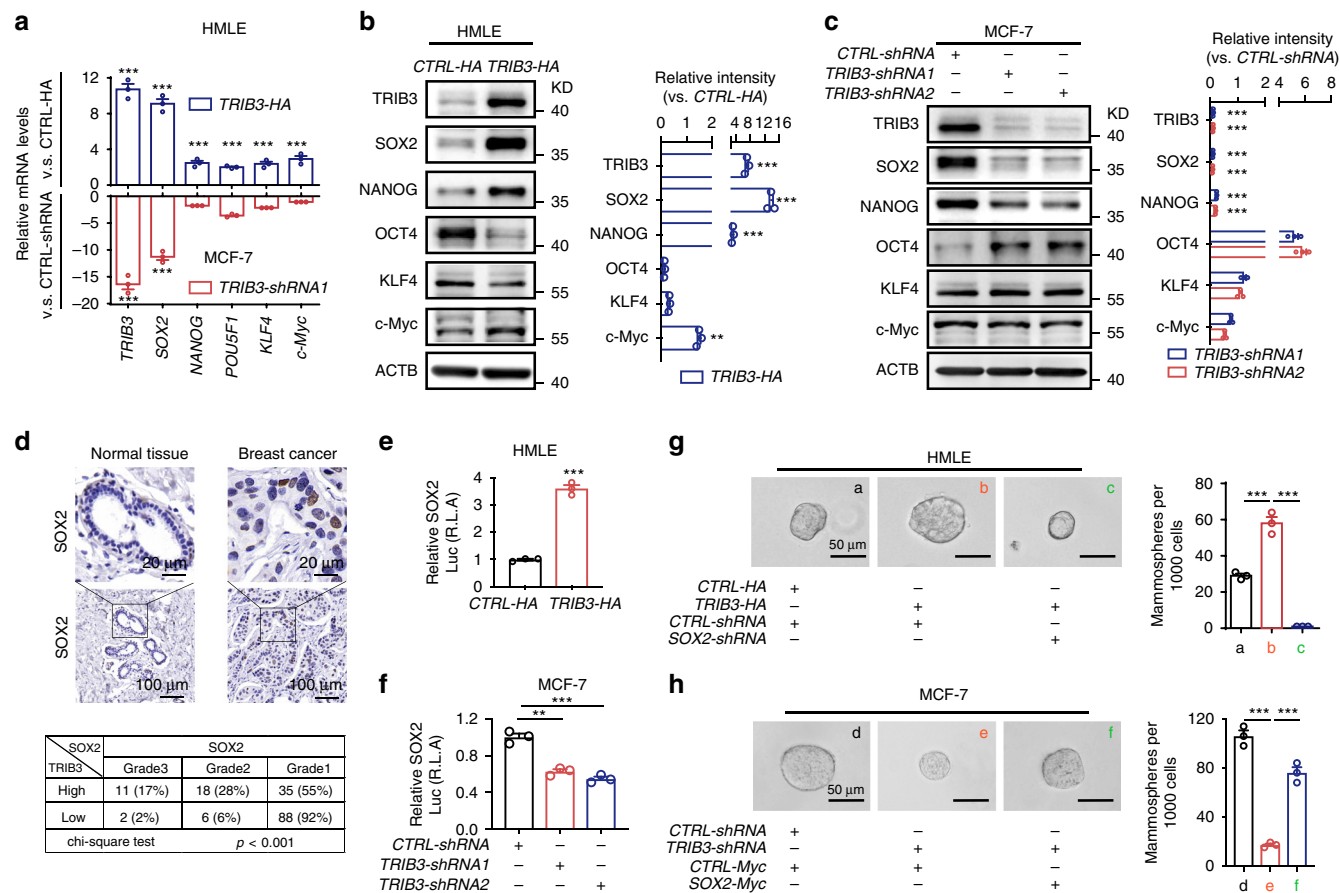

**Fig. 3 TRIB3 enhances SOX2 expression in BCCs. a** Histogram showing the expression of *TRIB3* (far left) and stem cell-related genes in HMLE cells transfected with *TRIB3*-expressing vector vs. control vector and MCF7 cells transfected with *TRIB3-shRNA1* vs. *CTRL-shRNA*. **b**, **c** Immunoblots and statistical analysis ($n = 3$) of protein lysates, as indicated on the left, transfected with a control vector or a *TRIB3*-expressing vector for HMLE cells (**b**) or with *CTRL-shRNA*, or *TRIB3-shRNA* for MCF7 cells (**c**). **d** Formalin-fixed, paraffin-embedded tissue microarray sections of normal and breast cancer tissues were stained with an anti-SOX2 antibody. Tissue-bound SOX2 is shown in brown (top). A 3 × 2 contingency table shows the correlation between TRIB3 and SOX2 based on their relative expression, graded by the percentage and intensity of the brown color staining in the immunohistological images (bottom). **e** Overexpression of *TRIB3* enhanced the SOX2 transcriptional activity in HMLE cells, as determined by a luciferase reporter assay. **f** Silencing TRIB3 inhibited the SOX2 transcription activity in MCF7 cells, as determined by a luciferase reporter assay. **g**, **h** Tumor spheroid formation ability of HMLE cells (**g**) transfected with a control vector or a TRIB3-expressing vector and either *CTRL-shRNA* or *SOX2-shRNA* (as indicated at the bottom) or of MCF7 cells (**h**) transfected with *CTRL-shRNA* or *TRIB3-shRNA*, and either a control vector or a SOX2-expressing vector. All data are shown as the number of mammospheres per 1000 cells. Data are presented as the mean ± SEM of three independent assays; $P > 0.05$ was considered not significant (NS); *$P < 0.05$, **$P < 0.01$, and ***$P < 0.001$ compared with the vector or *CTRL-shRNA* group. Source data are provided as a Source Data file.

factors reduced tumorsphere formation in HMLE cells (Supplementary Fig. 3c top). In HMLE-TRIB3 cells, silencing anyone of *SOX2*, *NANOG*, *c-Myc* could reduce tumorspheres; however, silencing only SOX2 completely blocked TRIB3-promoted tumorsphere formation (Supplementary Fig. 3c top). Silencing *TRIB3* have additional effect in *NANOG-shRNA*, *OCT4-shRNA*, *KLF4-shRNA*, *c-Myc-shRNA* MCF7 cells, but not in *SOX2-shRNA* MCF7 cells (Supplementary Fig. 3c bottom). These results indicated that SOX2 is a major factor that mediates TRIB3-supporting breast cancer stemness. We thus focused on the effect of the TRIB3-SOX2 axis on breast cancer stemness.

Using human tissue microarrays of breast adenocarcinoma, SOX2 expression was determined to be higher in tumor tissues than in adjacent nontumor tissues, and the expression level of TRIB3 correlated with the expression level of SOX2 in tumor tissues (Fig. 3d). TRIB3 enhanced the SOX2 transcriptional activity in HMLE cells (Fig. 3e) and MCF7 cells (Fig. 3f) but did not change the protein stability of SOX2 (Supplementary Fig. 3d). Furthermore, silencing SOX2 expression reduced the capacity of

HMLE cells transfected with a TRIB3-expressing vector to form spheroids (Fig. 3g). Overexpression of SOX2 rescued the spheroid formation capacity of TRIB3-silenced MCF7 (Fig. 3h) and MDA-231 cells (Supplementary Fig. 3e). Silencing SOX2 had no additional effects in terms of reducing of $CD44^{Br}CD24^{Di}$ subpopulation and limiting tumorsphere formation in *TRIB3-shRNA* MCF7 cells (Supplementary Fig. 3f, g). These data suggest that elevated SOX2 expression is responsible for TRIB3-supported breast cancer stemness.

To examine the dominant transcription factor (TF) that controls SOX2 expression in BCCs, we screened and silenced nine different SOX2-promoting TFs one by one in MCF7 cells. We found that silencing FOXO1 expression (Fig. 4a) but not that of any other TF (Supplementary Fig. 4a) reduced SOX2 expression in BCCs. Silencing FOXO1 expression reduced the SOX2 transcriptional activity in HMLE cells transfected with a TRIB3-expressing vector (Fig. 4b). Overexpression of FOXO1 rescued the SOX2 transcription activity in TRIB3-silenced MCF7 cells (Fig. 4c). Moreover, we identified the *SOX2* promoter-binding

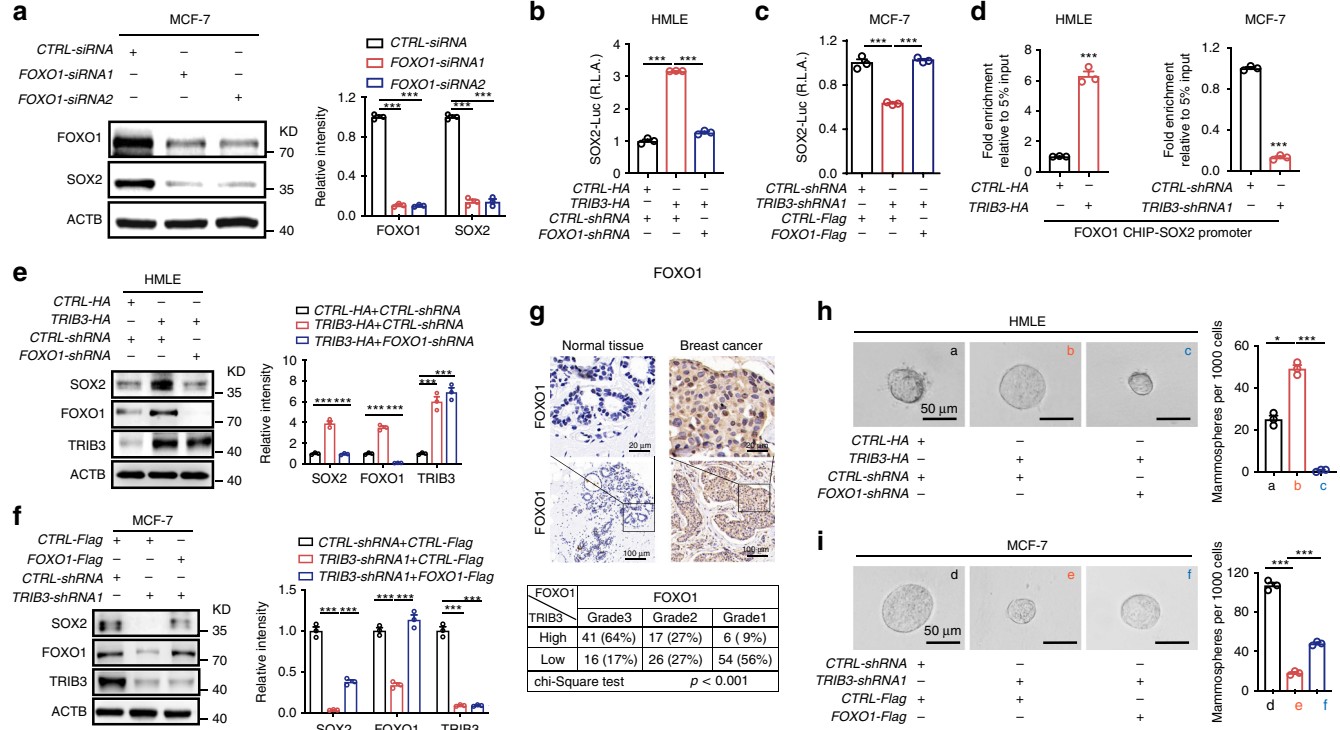

**Fig. 4 FOXO1 controls the TRIB3-SOX2 Axle in BCCs. a** Sample immunoblots and quantitative analyses of the expression of SOX2 after silencing the SOX2 transcriptional factor FOXO1 in MCF7 cells. **b**, **c** SOX2 transcriptional activity in HMLE cells (**b**) transfected with a control vector or a TRIB3-expressing vector and either *CTRL-shRNA* or FOXO1-*shRNA* (as indicated at the bottom), or in MCF7 cells (**c**) transfected with *CTRL-shRNA* or *TRIB3-shRNA*, and either a control vector or a FOXO1-expressing vector. **d** FOXO1 ChIP-SOX2 promoter activity in HMLE cells transfected with a control vector or a TRIB3-expressing vector (as indicated at the bottom) or MCF7 cells transfected with either *CTRL-shRNA* or *TRIB3-shRNA*. **e**, **f** Immunoblots and statistical analysis of protein lysates of HMLE cells (**e**) transfected with a control vector or a TRIB3-expressing vector and either *CTRL-shRNA* or *FOXO1-shRNA* (as indicated on the top), or of MCF7 cells (**f**) transfected with *CTRL-shRNA* or *TRIB3-shRNA* and either a control vector or a FOXO1-expressing vector. **g** Formalin-fixed, paraffin-embedded tissue microarray sections of normal or breast cancer tissues were stained with an anti-FOXO1 antibody. Tissue-bound FOXO1 is shown in brown (top). A 3 × 2 contingency table shows the correlation between TRIB3 and FOXO1 based on their relative expression, graded by the percentage and intensity of the brown color staining in the immunohistological images (bottom). **h**, **i** Tumor spheroid formation of HMLE cells (**h**) transfected with a control vector or a TRIB3-expressing vector and either *CTRL-shRNA* or *FOXO1-shRNA* (as indicated at the bottom), or of MCF7 cells (**i**) transfected with *CTRL-shRNA* or *TRIB3-shRNA* and either a control vector or a FOXO1-expressing vector. All data are shown as the number of mammospheres per 1000 cells. Data are shown as the mean ± SEM; $P > 0.05$ was considered not significant (NS); *$P < 0.05$, **$P < 0.01$, and ***$P < 0.001$ compared with the vector or *CTRL-shRNA* group. Source data are provided as a Source Data file.

activity of FOXO1 in HMLE, MCF7 (Fig. 4d), and MDA-MB-231 (Supplementary Fig. 4b) cells by chromatin immunoprecipitation (ChIP) assays. Silencing FOXO1 expression reduced SOX2 expression in HMLE cells transfected with a TRIB3-expressing vector (Fig. 4e). Overexpression of FOXO1 rescued SOX2 expression in TRIB3-silenced MCF7 cells (Fig. 4f) and MDA-MB-231 cells (Supplementary Fig. 4c). These data indicate that FOXO1 enhances SOX2 transcriptional expression in TRIB3-overexpressed BCCs.

Moreover, overexpression of TRIB3 enhanced FOXO1 expression and SOX2 nuclear accumulation in HMLE cells (Fig. 4e and Supplementary Fig. 4d). Silencing TRIB3 expression reduced FOXO1 accumulation in MCF7 (Fig. 4f and Supplementary Fig. 4d) and MDA-MB-231 cells (Supplementary Fig. 4c, d), and inhibited nuclear SOX2 expression in MCF7 and MDA-MB-231 cells (Supplementary Fig. 4d). The grade of FOXO1 expression was correlated with the TRIB3 expression in tumor tissues based on immunohistochemistry analysis of human tumor tissue microarrays of breast adenocarcinoma (Fig. 4g). Conversely, silencing FOXO1 expression reduced the spheroid formation of HMLE cells overexpressing TRIB3 (Fig. 4h). Overexpression of FOXO1 rescued the spheroid formation capacity of TRIB3-silenced MCF7 (Fig. 4i) and MDA-MB-231 cells (Supplementary

Fig. 4e). The FOXO1+SOXO2+ BCCs were identified by primary and PDX BCCs (Supplementary Fig 4f–g). These data indicate that FOXO1 coordinates with SOX2 to support the stemness of TRIB3-overexpressed BCCs.

**TRIB3 enhances FOXO1 expression**. We next investigated how TRIB3 upregulates FOXO1 expression in BCCs. TRIB3 markedly enhanced the half-life of FOXO1 degradation from 3 to 6 h in HMLE cells (Fig. 5a). Silencing TRIB3 reduced the half-life of FOXO1 degradation from 3.1 to 0.8 h (Fig. 5b). The degradation of FOXO1 was inhibited by the proteasome inhibitor MG132 but not by the autophagy inhibitor bafilomycin (BAF) in MCF7 cells (Fig. 5c). Silencing TRIB3 reduced FOXO1 expression in MCF7 and MDA-MB-231 cells, which was inhibited by MG132 but not by BAF (Fig. 5d). These data suggest that TRIB3 interferes with FOXO1 degradation by compromising the ubiquitin-proteasome system. The activation of various protein kinases often leads to the phosphorylation and ubiquitination of FOXO proteins[24]. Overexpression of TRIB3 reduced the S256, T24, and S319 phosphorylation of FOXO1 in HMLE cells (Fig. 5e). Silencing TRIB3 enhanced FOXO1 phosphorylation at S256, S319, but not at T24 in MCF7 cells (Fig. 5f). Indeed, ectopic expression of

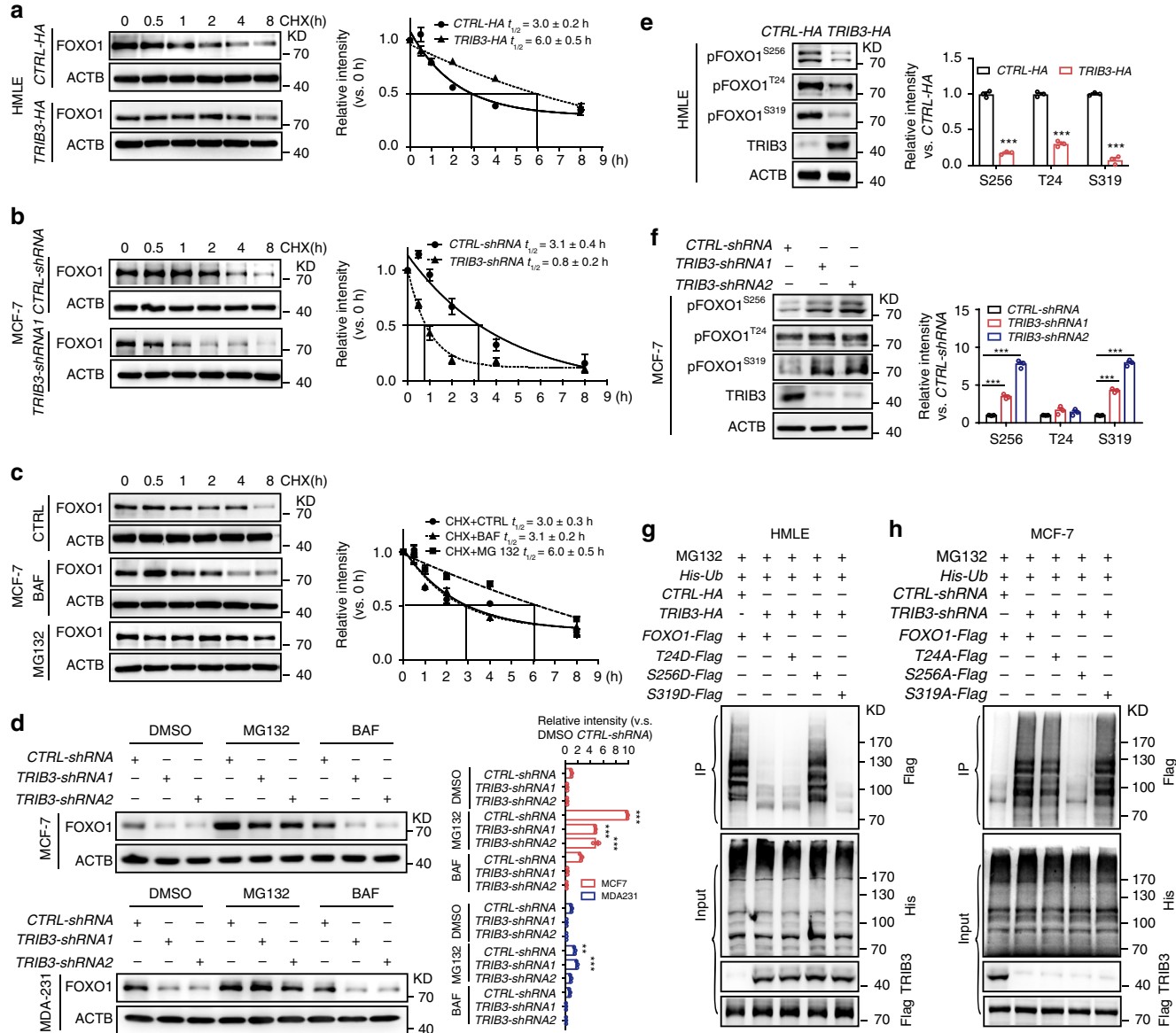

**Fig. 5 TRIB3 stabilizes the FOXO1 protein by suppressing its ubiquitination. a**, **b** HMLE cells transfected with a *TRIB3*-expressing vector vs. a control vector (**a**) or MCF7 cells transfected with *TRIB3-shRNA* vs. *CTRL-shRNA* (**b**) were incubated with 20 μM cycloheximide (CHX) for the indicated times. **c**, **d** The effect of TRIB3 depletion on FOXO1 degradation in vitro. MCF7 cells (**c**) were incubated with CHX (20 μM), CHX plus bafilomycin (BAF, 200 nM), or MG132 (10 μM) for the indicated times. Control or TRIB3-silenced MCF7 cells (**d**, top) and MDA-MB-231 cells (**d**, bottom) were incubated with DMSO, 200 nM BAF, or 10 μM MG132 for 8 h. The indicated proteins were detected by immunoblotting. **e**, **f** FOXO1 activity in HMLE cells (**e**) transfected with a control vector or a TRIB3-expressing vector, or in MCF7 cells (**f**) transfected with *CTRL-shRNA* or *TRIB3-shRNA* and either a control vector or a FOXO1-expressing vector. **g**, **h** The effect of TRIB3 on FOXO1 ubiquitination. HMLE (**g**) or MCF7 cells (**h**) were transfected with the indicated plasmid. His-ubiquitin-conjugated proteins were pulled down from cell lysates. Total protein lysates and Ni-NTA-agarose eluates were immunoblotted for FLAG, His, and TRIB3. Source data are provided as a Source Data file.

TRIB3 suppressed FOXO1 ubiquitination but did not reduce the ubiquitination of the S256D FOXO1 mutant in HMLE cells (Fig. 5g). Silencing TRIB3 enhanced FOXO1 ubiquitination but not S256A-mutated FOXO1 ubiquitination in MCF7 cells (Fig. 5h). These data suggest that phosphorylation of FOXO1 at S256 is a key signal for FOXO1 ubiquitination and degradation.

To determine how TRIB3 stabilized FOXO1 protein in BCCs, we analyzed the microarray transcriptome of TRIB3-silenced MCF7 cells. The expression of 91 FOXO-related E3 ligase genes was identified by TRIB3 silencing and *CBLL1*, *NEDD4L*, *UBE3C*, and *SKP2* were significantly elevated (Supplementary Fig. 5a). Silencing *TRIB3* enhanced protein expression of NEDD4L but not UBE3C, CBLL1, or SKP2 (Supplementary Fig. 5b). Silencing

NEDD4L and SKP2 but not UBE3C, CBLL1, or deubiquitinating enzymes USP5, USP14, and USP32 increased FOXO1 expression in MCF7 cells (Supplementary Fig. 5c, d). As we identified that phosphorylation of FOXO1 at S256 was a key signal for FOXO1 ubiquitination in BCCs, we examined the ubiquitination of the FOXO1-S256D-mutated form. SKP2 (Supplementary Fig. 5e) but not NEDD4L (Supplementary Fig. 5f, g) was found to mediate FOXO1 degradation by S256 phosphorylation. NEDD4L is the only E3 ligase whose expression can be reduced by TRIB3. We found that NEDD4L interacted with the FOXO1 C terminus and mediated FOXO1 ubiquitination (Supplementary Fig. 6a, b). K354, K446, K463, and K559 mediated FOXO1 ubiquitination in HMLE cells (Supplementary Fig. 6c). K463 was the key site at

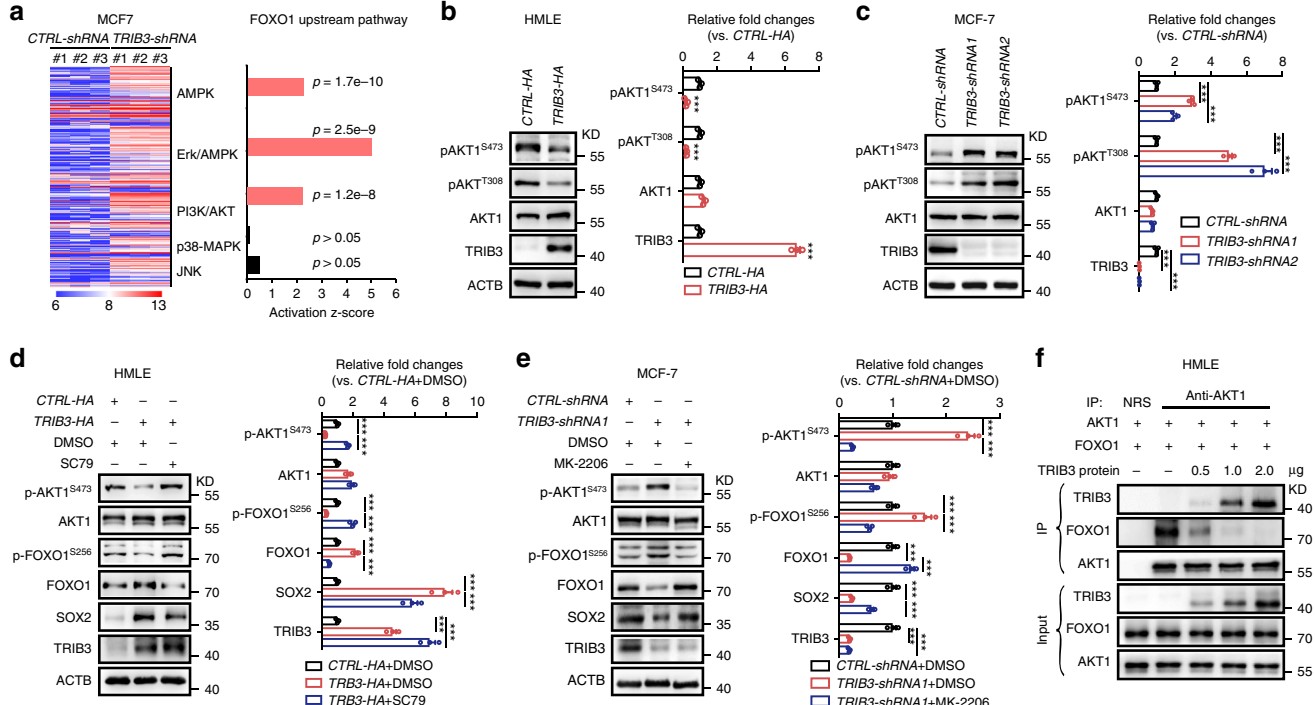

**Fig. 6 TRIB3 abrogates the AKT-FOXO1 interaction. a** Supervised clustering of MCF7 cells transfected with *CTRL-shRNA* or *TRIB3-shRNA* using the log2-transformed values for genes in the FOXO1 upstream pathway, including AMPK, ERK, PI3K, p38, and JNK clustered genes expressed in all samples. Each column represents a separate case. *Z*-scores from IPA depicting the subnetwork gene expression differences between *CTRL-shRNA* and *TRIB3-shRNA* cases. The *Z*-score heat map depicts the five FOXO1 upstream subnetworks. Each row represents the data for the subnetwork indicated in the right margin. **b**, **c** TRIB3 inhibited the activity of AKT1 in BCCs. The indicated stable expression cell extracts were prepared and the levels of the indicated proteins were detected by immunoblotting. **d**, **e** The effects of the AKT1 agonist or antagonist for TRIB3-FOXO1-SOX2 axis in BCCs. The indicated cells were treated with the AKT1 agonist SC79 or the antagonist MK2206 for 24 h. Cell extracts were prepared and the levels of the indicated proteins were detected by immunoblotting. **f** TRIB3 inhibited the AKT1-FOXO1 interaction by binding to AKT1. HMLE cell extracts were IP with an anti-AKT1 Ab and blotted with an anti-TRIB3 or anti-FOXO1 Ab in the presence of the indicated concentration of the TRIB3-GST protein. Data are shown as the mean ± SEM; $P > 0.05$ was considered not significant (NS); *$P < 0.05$, **$P < 0.01$, and ***$P < 0.001$ compared with the vector or *CTRL-shRNA* group. Source data are provided as a Source Data file.

which NEDD4L mediated FOXO1 ubiquitination (Supplementary Fig. 6d–f). Overexpression of FOXO1 increased tumorsphere formation in TRIB3-silenced BCCs. S256A, K463R, and S256A-K463A mutant FOXO1 further enhanced tumorsphere formation in TRIB3-silenced MCF7 cells (Supplementary Fig. 6g), indicating that SKP2 and NEDD4L contribute to TRIB3-mediated FOXO1 accumulation.

We utilized the same microarray transcriptome in TRIB3-silenced MCF7 cells to analyze the upstream signals of FOXO1 in BCCs. The genes of FOXO1-related pathways were listed in five sections, including AMPK, Erk-AMPK, PI3K, p38-MAPK, and JNK (Fig. 6a and Supplementary Fig. 7a). Subnetwork analyses revealed three subnetworks that were expressed at different levels between the TRIB3-silenced and control MCF7 cells. TRIB3-silenced MCF7 cells had higher-level subnetwork activity associated with AMPK, Erk/AMPK, and PI3K-AKT (Fig. 6a). We then investigated the expression of p-AMPK$^{T172}$, p-ERK$^{T204,T202}$, p-AKT$^{S473,T308}$, p-JNK$^{T183,185}$, and p-p38$^{T180,T182}$ in TRIB3-enhanced HMLE cells (Fig. 6b and Supplementary Fig. 7c), TRIB3-silenced MCF7 (Fig. 6c and Supplementary Fig. 7c), and MDA-MB-231 cells (Supplementary Fig. 7b, c). The reduced activities of AKT$^{S473,T308}$, AMPK$^{T172}$, and JNK$^{T183, T185}$ were found in TRIB3-overexpressed HMLE cells. Silencing TRIB3 activated AKT$^{S473, T308}$ and AMPK$^{T172}$ in MCF7 cells and activated AKT $^{S473, T308}$, JNK$^{T183, T185}$, and p38$^{T180, T182}$ in MDA-MB-231 cells (Fig. 6b, c and Supplementary Fig. 7b, c). Thus, TRIB3 consistently reduced the phosphorylation of AKT1, a kinase that can interact with FOXO1, as reported previously[25],

indicating that the AKT-FOXO1 interaction plays a critical role in TRIB3-enhanced breast cancer stemness. Indeed, we found that the AKT1 agonist SC79 rescued the activity of AKT1$^{S473}$ and FOXO1$^{S256}$, and inhibited FOXO1 and SOX2 expression in HMLE cells overexpressing TRIB3 (Fig. 6d). The AKT1 inhibitor MK2206 suppressed the activity of AKT1$^{S473}$ and FOXO1$^{S256}$, and rescued FOXO1 and SOX2 expression in TRIB3-silenced MCF7 cells (Fig. 6e) and MDA-MB-231 cells (Supplementary Fig. 7d). Silencing *AKT1* also rescued FOXO1 and SOX2 expression in TRIB3-silenced MCF7 cells (Supplementary Fig. 7e). Stress factors including IL-6, hypoxia, and high glucose reduced the activity of FOXO1$^{S256}$ and enhanced expressions of FOXO1 and SOX2 in MCF7 cells (Supplementary Fig. 7f). It was reported that *TRIB3* is a direct FOXO1 target gene in PC12 mouse neurons[26]. Overexpression of FOXO1 enhanced TRIB3 expression in MCF7 cells (Supplementary Fig. 7g). However, hypoxia, glucose deprivation, and IL-6 could enhance *TRIB3* expression in FOXO1-silenced BCCs (Supplementary Fig 7h). These data indicate that FOXO1 is not the only nuclear TF for stress protein TRIB3 and FOXO1-induced TRIB3 expression provides a feedforward regulation of the TRIB3-FOXO1-SOX2-mediated breast cancer stemness. Moreover, TRIB3 did not endogenously interact with FOXO1 in MCF7 cells (Supplementary Fig. 7i) but did dose-dependently abrogate the AKT1-FOXO1 interaction in HMLE cells (Fig. 6f).

On the other hand, overexpression of TRIB3 in HMLE cells enhanced the mRNA expression of *FOXO1* (Supplementary Fig. 8a) and *FOXO1* luciferase activity (Supplementary Fig. 8b).

Silencing TRIB3 attenuated the mRNA expression of *FOXO1* and *FOXO1* luciferase activity in MCF7 cells (Supplementary Fig. 8a, b). Interestingly, silencing SOX2 reduced the transcriptional activity of FOXO1 in HMLE cells transfected with a TRIB3-expressing vector (Supplementary Fig. 8c), but overexpression of SOX2 rescued the transcriptional activity of FOXO1 in TRIB3-silenced MCF7 cells (Supplementary Fig. 8c). SOX2 binds DNA in a sequence-specific manner via its high-mobility-group domain[27]. We analyzed a 5 kb locus upstream of the FOXO1 gene transcription start sites (TSS). Using ChIP, we found that the −246 ~ −1460 base pair (bp) region of the FOXO1 promoter upstream of the TSS was occupied by the SOX2 protein (Supplementary Fig. 8d). We evaluated whether FOXO1 transcriptional activity was promoted by SOX2 using luciferase reporter constructs. Based on these assays, a −1032 ~ −793 bp fragment of the FOXO1 promoter was identified as an enhancer region in which SOX2 was activated (Supplementary Fig. 8e–g). We then mapped the binding locus on the FOXO1 promoter using electrophoretic mobility shift assays (EMSAs) with different oligos from the −1032 ~ −753 bp fragment. The −882 ~ −843 bp segment of the FOXO1 promoter was found to contain the SOX2-binding sequence (Supplementary Fig. 8h–j). These data suggest that a positive feedback mechanism of the TRIB3-FOXO1-SOX2 axis supports breast cancer stemness. TRIB3 enhances FOXO1 expression by impairing its degradation, the elevated FOXO1 induces SOX2 transcriptional expression, and the latter conversely activates FOXO1 transcriptional expression.

### Disturbing the TRIB3/AKT interaction reduces cancer stemness.
We examined whether TRIB3 reduced AKT1 phosphorylation by interacting with AKT1. The endogenous or overexpressed AKT1 was co-immunoprecipitated (IP) with TRIB3. The deletion mutants of Myc-tagged AKT1 were subjected to IP with TRIB3-HA. TRIB3 interacted with the region from residues 149 to 214 in the catalytic domain of AKT1 (Fig. 7a–c). Furthermore, the deletion mutants of green fluorescent protein (GFP)-tagged TRIB3 were used to map the AKT1 interaction region (Fig. 7d). AKT1 interacted with the C terminus of the TRIB3 KD domain (Fig. 7e).

To verify the critical role of the TRIB3/AKT1 interaction in the regulation of FOXO1 expression and tumor stemness, we tried to interrupt this interaction by screening for a short α-helical peptide able to inhibit this protein–protein interaction (PPI)[14,15]. We first docked the TRIB3 protein, created by homology models, with the AKT1 protein from Protein Data Bank (PDB, 4GV1) by Discovery Studio. The α-helical region at residue 191–204, which is in the AKT1 catalytic domain, is the closest α-helical region in the TRIB3-AKT1 interaction range (Fig. 7f, yellow α-helix). We further screened the abilities of a series of α-helical peptides to disturb the TRIB3/AKT1 interaction based on the crystallized secondary structure of the AKT1 catalytic domain available in the PDB (Fig. 7f, AKT1 red regions). We found that peptide Ae, which mimics the AKT1 191–204 α-helical region (Fig. 7f, yellow α-helix), displayed the best binding affinity with TRIB3 among all the α-helical peptides analyzed (Fig. 7g). The AKT1 230–325 region (WX region) reportedly mediates the TRIB3/AKT1 interaction. We further identified that deletion of the Ae mutant (△Ae) but not deletion of the WX mutant (△WX, 230-315AA) inhibited the binding of AKT1 to TRIB3 (Fig. 7h, i). The TRIB3-binding regions of peptide Ae were predicted to reside inside the TRIB3 KD domain (Fig. 7j). These data indicate that the catalytic region of AKT1 (residues 149−214, especially 191−204) interacts with the C terminus of the TRIB3 KD domain to cause downstream effects. To assess the contribution of the peptide Ae amino-acid sequence to the TRIB3/AKT1 binding, each

amino-acid residue of Ae was substituted with alanine. The residues 1V(M1), 5L(M5), 6T(M6), 11L(M11), 12Q(M12), and 13N(M13) of peptide Ae were critical for the binding of Ae to TRIB3, because the alanine mutations abolished Ae/TRB3 binding (Supplementary Fig. 9a).

To verify whether peptide Ae interrupted the TRIB3/AKT1 interaction in BCCs, a fused peptide, Pep2–Ae, was designed by linking a cell-penetrating peptide to Ae using a glycine–glycine linker[14]. Pep2–Ae exerted specific binding effects for TIRB3/AKT1 interaction. SMAD3, P62, and β-catenin were found to interact with TRIB3[14,28,29]. Pep2–Ae neither had effects on TRIB3 interactions with SMAD3, P62, or β-catenin (Supplementary Fig. 9b), nor interfered with AKT1 interacted proteins including PDK1, GSK3β, and P85 (Supplementary Fig. 9c). Pep2–Ae dose-dependently inhibited the TRIB3/AKT1 interaction in MCF7 and MDA-MB-231 cells (Supplementary Fig. 10a). Also, Pep2–Ae enhanced the phosphorylation of AKT1 and FOXO1, reduced SOX2 and FOXO1 expression in MCF7 and MDA-231 cells (Supplementary Fig. 10b), inhibited FOXO1 and SOX2 transcriptional activity (Supplementary Fig. 10c), and accelerated FOXO1 degradation from 2.6 to 0.6 h in MCF7 cells (Supplementary Fig. 10d). Thus, Pep2–Ae reduced tumor mammosphere formation in MCF7 (Supplementary Fig. 10e) and MDA-231 cells (Supplementary Fig. 10f).

Pep2–Ae reduced the mammosphere formation ability of MECs from MMTV-PyMT (Fig. 8a) and MMTV-ErbB2 (Fig. 8b) transgenic mice and primary BCCs from PDX mice (Fig. 8c). MECs from MMTV-PyMT transgenic mice (Fig. 8d top) and primary BCCs from PDX mice (Fig. 8d bottom) treated with Pep2–Ae displayed no significant effect on primary tumor growth (Supplementary Fig. 10g, h), but a reduced capacity for secondary tumor formation in FVB mice (Fig. 8d top) and immunodeficient mice (Fig. 8d bottom). Indeed, the TRIB3/AKT1 interaction was abrogated in MMTV-PyMT MEC-derived breast tumors and PDX breast tumors (Fig. 8e). Finally, Pep2–Ae enhanced the activity of AKT1 and FOXO1, and reduced the expression of FOXO1 and SOX2 in MMTV-PyMT MEC-derived breast tumors and PDX-derived breast tumors (Fig. 8f). These data verify that interrupting the TRIB3/AKT1 interaction reduces breast cancer stemness and exerts antitumor efficacy by promoting FOXO1 phosphorylation, ubiquitination, and degradation, which suppresses SOX2 expression (Fig. 8g).

## DISSCUSION
The double threat of CSCs includes therapy resistance and the ability to regenerate a tumor from original and distant sites once therapy is halted[4]. Many tumors, including breast cancer, harbor CSCs in dedicated niches[6,30]. The CSC niches or microenvironments determine not only the tumor progression molecular heterogeneity but also the fate of CSCs[31]. Both intrinsic and extrinsic stressors, including hypoxia, metabolism, reactive oxygen species, and inflammation, upregulate the CSC stress signaling pathway to enhance cancer cell survival and maintain cancer cell stemness[32]. However, the molecular mechanisms by which the stress signaling pathway sustains cancer cell stemness remain unclear. TRIB3 can act as a stress sensor in response to all of these stressors and thus participates in the pathogenesis of chronic inflammatory and malignant diseases by interacting with intracellular signaling and functional proteins[14,33–35]. In this study, we demonstrated that elevated expression of TRIB3 is positively associated with the initiation and progression of breast cancer as well as with the poor prognosis of breast cancer patients by enhancing breast cancer stemness. TRIB3 protein acts as a stress sensor in response to a diverse range of stressors, including inflammation, hypoxia, and metabolic stress in BCCs. Enhanced

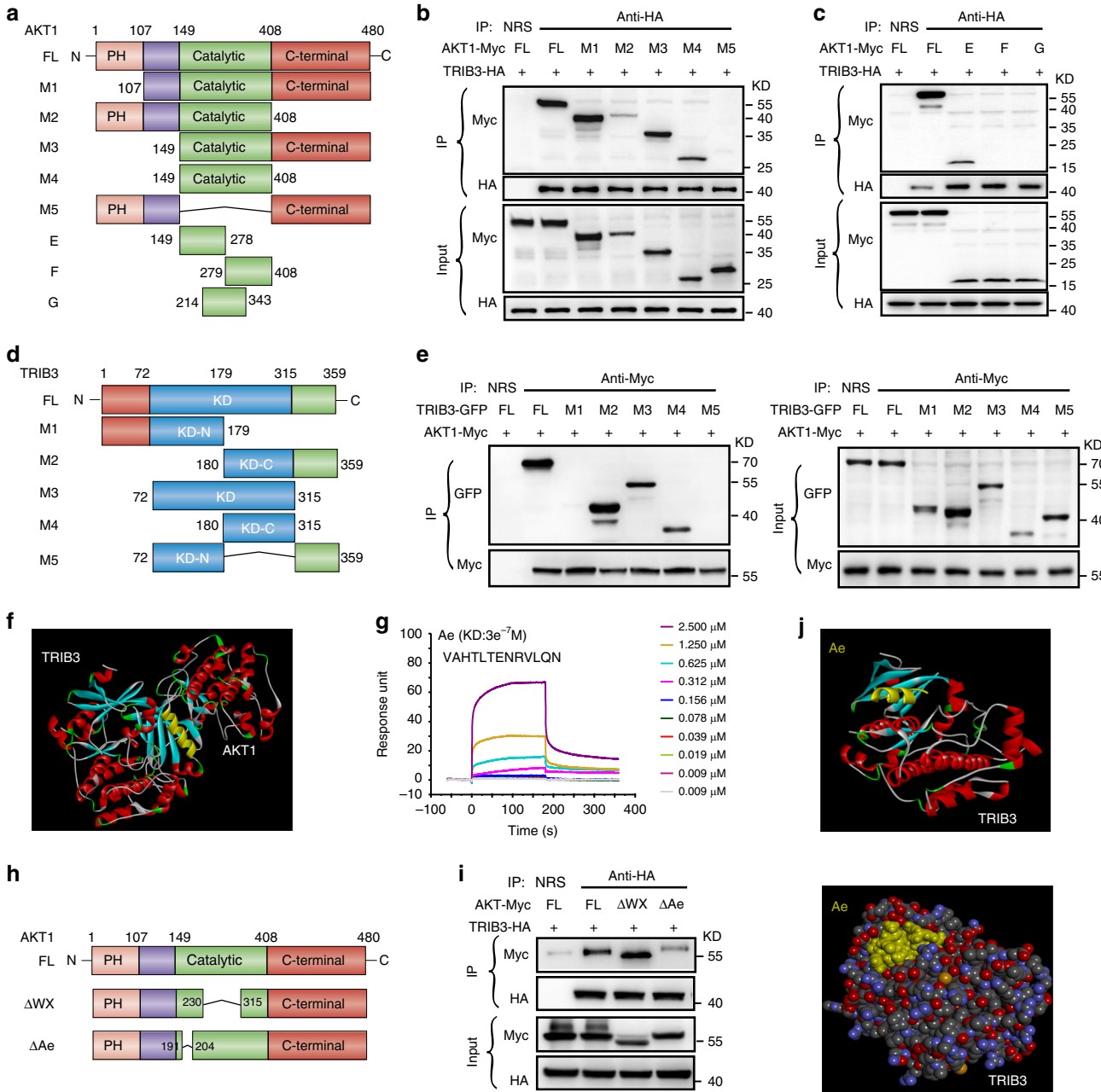

**Fig. 7 Mapping of the TRIB3-AKT1 interaction. a** Mapping AKT1 regions binding to TRIB3. Schematic diagram of full-length AKT1 and deletion mutants. **b**, **c** HEK293T cells were co-transfected with the indicated *AKT1*-Myc and *TRIB3*-HA constructs. Cell extracts were IP with an anti-HA Ab **d** Mapping TRIB3 regions binding to AKT1. Schematic diagram of full-length TRIB3 and deletion mutants. **e** HEK293T cells were co-transfected with the indicated AKT1 and TRIB3-GFP constructs. Cell extracts were IP with an anti-Myc Ab. **f** Prediction of the AKT1 and TRIB3 interaction using Discovery Studio. **g** Kinetic interactions of α-helical peptides and the TRIB3 protein were determined by SPR analyses. **h** Schematic diagram of full-length AKT1 and deletion mutants. **i** The △Ae *AKT1* mutant barely binds the TRIB3 protein. HEK293T cells were co-transfected with the indicated mutants of *AKT1*-Myc and *TRIB3*-HA. Cell extracts were IP with an anti-HA Ab. **j** Prediction of the Pep2−Ae and TRIB3 interaction by Discovery Studio. Source data are provided as a Source Data file.

TRIB3 activates the AKT1-FOXO1-SOX2 axis in TRIB3-overexpressed BCCs. Importantly, interrupting the TRIB3/AKT1 interaction by a modified α-helix peptide, Pep2−Ae, accelerated FOXO1 degradation, reduced SOX2 accumulation, and decreased the tumor-initiating capacity in MMTV-PyMT and PDX mice. Our work indicates that TRIB3 links stress signals to breast cancer stemness through the coordination of FOXO1 and SOX2 in breast cancer with high TRIB3 expression.

Overexpression of pluripotency genes, such as *SOX2*, *OCT4*, *NANOG*, *KLF4*, and *c-Myc*, can induce somatic cells to acquire pluripotency[36]. These proteins also show functional differences in

CSCs[37–39]. Evidences showed that the expression of a single stemness-contributing protein might have a relatively broad range[40,41]. In this study, we found that mammary CD24+CD29lo cells from MMTV-PyMT mice and CD24+CD29hi cells from MMTV-ErbB2 mice express the higher TRIB3 than their age-matched normal counterparts. Moreover, isolated MCF7 CD44BrCD24Di cells had higher TRIB3 and SOX2 expression levels than their counterparts. SOX2 plays an essential role, by not only regulating pluripotency but also mediating self-renewal and differentiation. SOX2 is aberrantly expressed in several types of cancers, such as breast, lung, ovarian and prostate cancers[41–45].

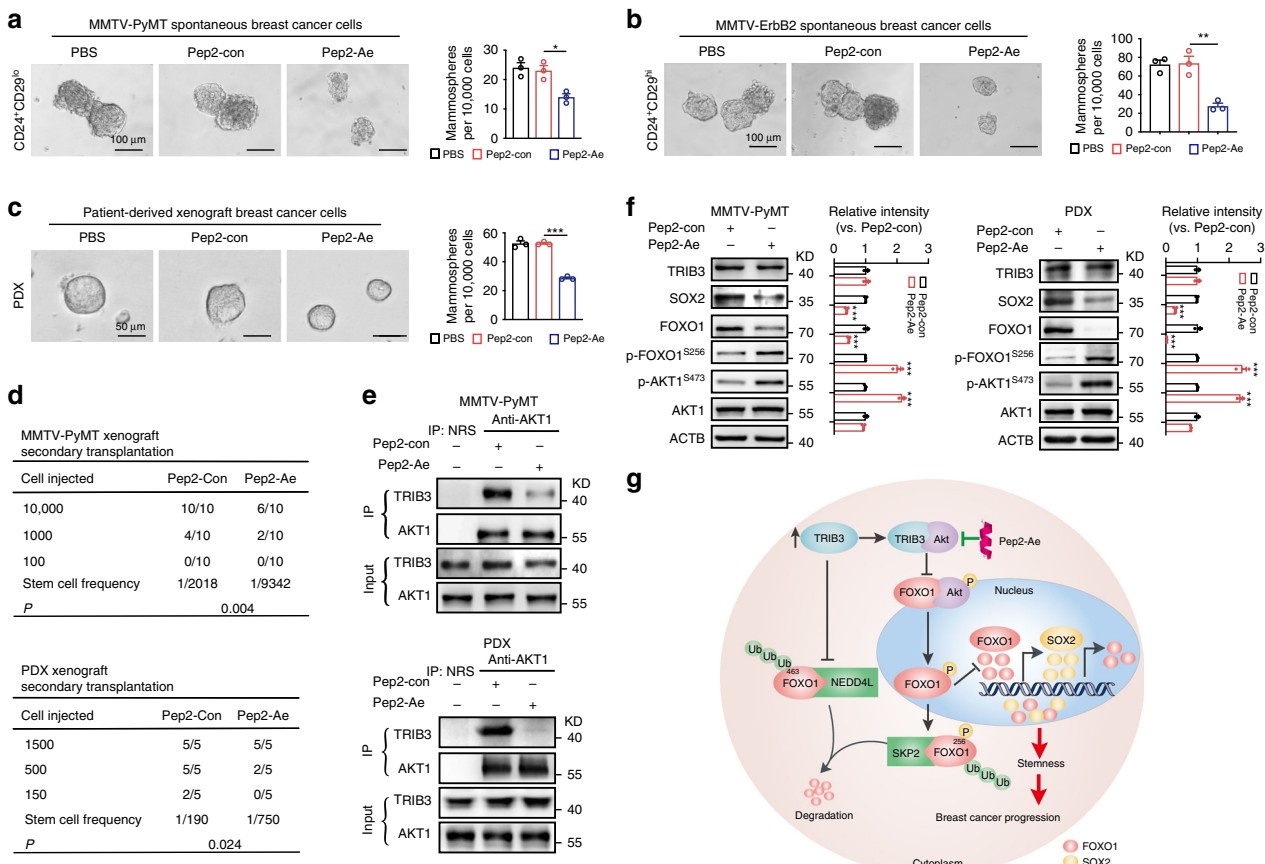

**Fig. 8 Pep2–Ae disturbs the TRIB3-AKT interaction and reduces breast cancer stemness. a–c** Pep2–Ae suppressed mammosphere formation. Primary MMTV-PyMT, MMTV-ErbB2, and PDX BCCs were cultured in tumorsphere media and treated with the indicated peptides (1 μM) for 5–7 days. The size and number of mammospheres were analyzed (scale bar, 100 μm or 50 μm). **d** The effects of Pep2–Ae treatment on the tumor incidence of secondary transplanted MMTV-PyMT-derived breast cancer in FVB/N mice (top) and PDX-derived BCCs in NPG mice (bottom). **e** The effect of Pep2–Ae on the TRIB3/AKT1 interaction. Primary MMTV-PyMT and PDX BCCs were isolated from peptide-treated engrafted mice. Cell extracts were IP with an anti-AKT1 Ab and blotted with an anti-TRIB3 Ab. **f** The effect of Pep2–Ae on the levels of the indicated proteins. Primary MMTV-PyMT and PDX BCCs were isolated from peptide-treated engrafted mice. Cell extracts were prepared, and the levels of the indicated proteins were detected by immunoblotting. **g** Schematic diagram illustrates the elevated TRIB3 expression promoted BCSCs and breast cancer development. Data are shown as the mean ± SEM; $P > 0.05$ was considered not significant (NS); *$P < 0.05$, **$P < 0.01$, and ***$P < 0.001$ compared with Pep2–con. Source data are provided as a Source Data file.

However, the mechanisms by which SOX2 and other tumor-initiating cell markers are overexpressed in cancer remain unclear. Recent studies demonstrated that SOX2-expressing cells are the founding CSC population driving tumor initiation and therapy resistance[46,47]. SOX2 expression has been shown to positively correlate with the cancer cell stemness of solid tumors, including breast cancer, and knockdown of SOX2 decreases invasiveness and cancer cell stemness[48–51]. The expression of SOX2 and SOX9 is essential for the survival and metastasis-initiating properties of latency-competent cancer cells in multiple host tissues[52]. In the current study, TRIB3-promoted tumor-sphere ability is reduced in SOX2- but not in c-MYC- or KLF4-silenced BCCs. Moreover, SOX2 but not c-MYC, NANOG, or KLF4 is consistently upregulated by TRIB3 in several different types of BCCs. We proposed that SOX2 rather than other pluripotency factors plays a vital role in supporting the breast cancer stemness in breast cancer with high TRIB3 expression. Indeed, nuclear accumulation of FOXO1, but not that of other TFs, promotes *SOX2* gene expression in breast cancer. FOXO1-dependent genes are implicated in stem cell renewal, migration and invasion, differentiation, and oxidative stress[53,54]. FOXO1, as a recently recognized pluripotency factor, coordinates with SOX2, to support stemness[54]. Elevated FOXO1 usually degrades quickly through ubiquitination pathway after phosphorylation in

physiological conditions. However, TRIB3-enhanced SOX2 can bind to the FOXO1 promoter, to further enhance *FOXO1* gene expression. This positive feedback loop causes FOXO1 nuclear accumulation and promotes breast cancer stemness. Moreover, inhibition of S256 FOXO1 phosphorylation by TRIB3 is responsible for FOXO1 accumulation. Further studies need to clarify potential additional pathways for CSC regulation other than SOX2 in TRIB3-elevated BCCs. For example, it remains unclear whether c-MYC, NANOG, or KLF4 play a role in maintaining stemness in a certain type of BCCs by using isolated CSCs based on multiple parameters.

Current CSC therapy includes the inhibition of key CSC pathways such as the WNT and NOTCH pathways, CSC ablation using antibody-drug conjugates, epigenetic therapy, and quiescent CSC eradication[43]. However, recent evidence, especially from research on CSC plasticity, suggests a need for rethinking that the removal of resident CSCs can cure cancer[43]. The approach to specifically target the interaction between CSCs and their niche or microenvironment rather than pursuing therapies based on intrinsic CSC features is important for improving patient outcomes. In the current study, the activity of AKT1 was universally regulated by TRIB3 in BCCs. The fact that the TRIB3-AKT1 interaction prohibits the AKT1-FOXO1 interaction provides an opportunity to modulate the effects of the niche-enriched stress

protein TRIB3, which indeed shows enhanced expression adjacent to CD68[+] TAMs in breast cancer patients. Several PPI modulators that inhibit the interactions among MDM2/p53, XIAP/caspase-9, and BCL2/beclin1 are being tested in clinical trials for cancer patients. The TRIB3/AKT1 interaction is reported to play roles in metabolic disease[55]. Here we demonstrated that Pep2–Ae, an α-helix from the catalytic region of AKT1, disrupts the TRIB3/AKT1 interaction and reduces the cancer initiation capacity of luminal and basal BCCs, and primary BCCs from transgenic mice and PDX mice via the activation of FOXO1 phosphorylation, ubiquitination, and degradation, which leads to the inhibition of SOX2 expression and breast cancer stemness. These results not only verify that the TRIB3-FOXO1-SOX2 signaling axis plays a key role in maintaining tumor initiation capacity, but also provide therapeutic options to target the interaction between the CSCs and their niche. Notably, we did not find that this AKT activation by TRIB3 blocking leads to tumor growth, which is consistent with the observation that AKT activation does not always lead to tumor proliferation[56]. However, further studies are needed to clarify the potential negative physiological signals for TRIB3-AKT-FOXO1-SOX2 loop and the exact role of AKT1 activation and upstream regulators such as PI3K by silencing TRIB3 expression.

In summary, our study indicates that the elevated stress protein TRIB3 links stress signals to induce breast cancer initiation and progression by supporting breast cancer stemness, which is coordinated with elevated FOXO1 and SOX2, and triggered by activation of the TRIB3-AKT1-FOXO1-SOX2 axis in TRIB3-overexpressed BCCs. Thus, this work provides a proof-of-concept for directly targeting BCSCs against breast cancer through inhibition of the TRIB3/AKT1 interaction.

## Methods

**Reagents**. FOXO1 (siRNA1-5′-CCAGATGCCTATACAACA-3′, siRNA2-5′-CA AATGCTAGTACTATTCT-3′), GLI2 (siRNA1-5′-CGTCAACCCTGTCGCC-3′, siRNA2-5′-GATCTGGACAGGGATGACT-3′), P27 (siRNA1-5′-CCGACGATT CTTCTACTCA-3′, siRNA2-5′-GGAGCAATGCGCAGGAATA-3′), FOXM1 (siRNA1-5′-CTCTTCTCCCTCAGATATA-3′, siRNA2-5′-CGGAAATGCTTGT GATTCA-3′), PAX6 (siRNA1-5′-GTGCGACATTTCCCGAATT-3′, siRNA2-5′-GCAGACGGCATGTATGATA-3′), FOXP1 (siRNA1-5′-CTGGTTCACACGA ATGTTT-3′, siRNA2-5′-CCACAGAGCTTACCTCATA-3′), GLI1 (siRNA1-5′-GCACCAAACGCTATACAGA-3′, siRNA2-5′-GCCTGAATCTGTGTATGA A-3′), CDKN1A (siRNA1-5′-CTTCGACTTTGTCACCGAG-3′, siRNA2-5′-GA CCATGTGGACCTGTCAC-3′) and SiRT1 (siRNA1-5′-GCTAAGAATTTCAG GATTA-3′, siRNA2-5′-GGAAATATATCCTGGACAA-3′) small interfering RNAs (siRNAs) and nontargeting (siN05815122147-1-5) siRNAs were purchased from Guangzhou RiboBio, Co., Ltd (China). All siRNA transfections were performed in Dulbecco's modified Eagle's medium (DMEM) serum-free medium using Lipofectaimine RNAiMAX (Invitrogen) according to the manufacturer's instructions. The AKT inhibitor MK2206 (S1078) and AKT activator SC79 (S786) were purchased from Selleck Chemicals (Houston, USA).

**Cell culture**. The BCC lines MCF7, T47D, BT474, MDA-MB-231, HS-578T, BT549, and MDA-MB-468 were obtained and validated with short tandem repeats analysis from National Infrastructure of Cell Line Resource, Peking Union Medical College (Beijing, China), and cultured in standard conditions according the American Type Culture Collection instructions[57]. All cell lines were verified negative for mycoplasma contamination by MycoAlert[TM] Mycoplasma Detection Kit (Lonza, LT07-418). No cells lines used here appear in the database of commonly misidentified cell lines (the International Cell Line Authentication Committee) except MDA-MB-435s, which was only used for the detection of TRIB3 expression in Fig. 1c. The immortalized HMLE cells were provided by Dr Guangli Suo (Suzhou Institute of Nano-Tech and Nano-Bionics (SINANO), Chinese Academy of Sciences). HMLE-Ras (HMLER) cells were generated by infecting the HMLE cells with lentivirus-H-RasV12 (HanBio, Co., Ltd, China), followed by selection with 2 μg/ml of puromycin (Gibco). HMLE and HMER cells were maintained in DMEM:F12 media (1:1) supplemented with 10% fetal bovine serum[58].

**Generation of stable cell lines**. Silencing TRIB3 was achieved by targeting the sequences 5′-CGGTTGGAGTTGGATGACAACTTAG-3′ (TRIB3-shRNA1) and 5′-GATCTCAAGCTGTGTCGCTTT-3′ (TRIB3-shRNA2) in the pHBLV-U6-

Scramble vector (HanBio, Co. Ltd, Beijing, China). Silencing SOX2 and FOXO1 was achieved with commercial SOX2-shRNA (sc-38408-V) and FOXO1-shRNA (sc-35382-V) lentivirus particles from Santa Cruz Biotechnology (TX, USA). Silencing SOX2, NANOG, POU5F1, MYC, and KLF4 was achieved by targeting the sequences 5′-CGCTCATGAAGAAGGATAAGT-3′ (SOX2-shRNA), 5′-GCTTTGAAGCATCCGACTGTA-3′ (NANOG-shRNA), 5′-CCCTCACTTCACT GCACTGTA-3′ (POU5F1-shRNA), 5′-CCTGAGACAGATCAGCAACAA-3′ (MYC-shRNA), and 5′-TCTCCTGGACCTGGACTTTAT-3′ (KLF4-shRNA) from Hanbio, Co. Silencing Trib3 was achieved by targeting the sequences 5′-CGGTTGGAGTTGGATGACAACTTAG-3′ (Trib3-shRNA1), 5′-GATCTCAA GCTGTGTCGCTTT-3′ (Trib3-shRNA2), and 5′-GATCTCAAGCTGTGTC GCTTT-3′ (Trib3-shRNA3) in the pHBLV-U6-Scramble vector (HanBio, Co.). Concentrate lentivirus particles were used to infect sub-confluent cultures in the presence of 5 μg/ml polybrene overnight. Twenty-four hours post transfection, cells were selected in media containing 2 μg/ml puromycin (Gibco).

HMLE-CTRL and HMLE-TRIB3 stable transfected cells were generated as follows. Human TRIB3 cDNA was cloned into the pcDNA3.1-HA vector and then transfected into HMLE cells using the Lipofectamine LTX&PLUS transfection reagent (Invitrogen). Stable cell lines were selected and cultured in DMEM with 0.8 mg/ml G418 (Gibco). SOX2 or FOXO1 overexpression was achieved by infection cells with SOX2-GFP or FOXO1-GFP lentivirus particles (HanBio, Co.), respectively. Stable SOX2- or FOXO1-overexpressed BCCs were selected by sorting GFP-positive cells by flow cytometry. Stable cell lines were cultured in DMEM with 0.4 mg/ml G418.

The efficiency of silencing or overexpression was validated by SYBR Green quantitative reverse transcriptase PCR (qRT-PCR) or immunoblot analysis with an anti-TRIB3 antibody (Abcam, ab75846), anti-SOX2 antibody (Cell Signaling Technology (CST), 3579), anti-FOXO1 antibody (CST, 2880), or anti-Trib3 antibody (Thermo Fisher Scientific, PA5–15480), which were used at a 1:1000 dilution. Glyceraldehyde 3-phosphate dehydrogenase (GAPDH) and β-Actin expression were used as endogenous controls for qRT-PCR and immunoblot analysis, respectively.

**Plasmid construction**. The following AKT1 truncations were constructed in the pcDNA3.1−myc-his vector via the BamHI and NheI restriction sites: full-length (FL amino acids 1–480), M1 (amino acids 107–480), M2 (amino acids 1–408), M3 (amino acids 149–480), M4 (amino acids 149–408), M5 (amino acids 1–149, 409 −480), E (amino acids 149–278), F (amino acids 279–408), and G (amino acids 214–343). The following TRIB3 truncations were constructed in the pEGFP-C1 vector via the BamHI and KpnI restriction sites: FL (amino acids 1–359), M1 (amino acids 1–179), M2 (amino acids 179–359), M3 (amino acids 72–315), M4 (amino acids 179–315), and M5 (amino acids 72–179, 315–359). The following FOXO1 promoter truncations were constructed in the pEZX-PG04 vector via the BglII and HindIII restriction sites: bases −4913 ~ −3863, bases −3887 ~ −2728, bases −2752 ~ −1866, bases −1892 ~ −1019, bases −1032 ~ −1, bases −793 ~ −1, bases −589 ~ −1, bases −326 ~ −1, and bases −50 ~ −1.

**Western blotting and protein stability assay**. Cycloheximide (CHX; Sigma Aldrich, R750107) was used as a protein synthesis inhibitor to assess the stability of the existing proteins. To assess protein stability, cells were treated with 20 μM CHX and the proteasome inhibitor MG132 (Sigma Aldrich, R750107) at 10 μM or the autophagy inhibitor BAF (Sigma Aldrich, B1793) at 200 nM for the indicated times. Protein expression was assessed via immunoblot analysis with cell lysates (20–60 μg) prepared in RIPA lysis buffer containing protease inhibitors (Roche) using anti-SOX2 (3529), anti-NANOG (4903), anti-OCT4 (2750), anti-KLF4 (4038), anti-c-Myc (13987), anti-FOXO1 (2880), anti-p-FOXO1 (9461), anti-AKT (4685), anti-p-AKT (4060), anti-β-ACTIN (4970), anti-Ubiquitin (3936), or anti-LaminA/C (4777) antibodies from CST; anti-TRIB3 antibody (ab75846) from Abcam; and anti-Trib3 antibody (PA5-15480) from Thermo Fisher Scientific[59]. All antibodies were used at 1:1000 dilution, with the exception of anti-β-Actin antibodies, which were used at 1:2000 dilution. The protein concentrations were determined by BCA Protein Assay Kits (Pierce). Proteins were separated by SDS-polyacrylamide gel electrophoresis (PAGE) and transferred to polyvinylidene difluoride membrane.

**Flow cytometry**. Mouse MECs were isolated from the fourth mammary glands of 6-week-old wild-type FVB mice, 6-week-old MMTV-PyMT transgenic mice, or 6-month-old MMTV-ErbB2 transgenic mice by using mouse tumor dissociation kits (Miltenyi Biotec, 130-096-730). MECs were incubated with biotin-conjugated lineage antibodies including anti-Cd31 (553371), anti-Cd45 (553078), and anti-Ter119 (553672) antibodies (1:100, BD Biosciences) for 15 min at 4 °C. MECs were then incubated with anti-biotin microbeads (Miltenyi Biotec, 130-090-485) for 15 min at 4 °C. Lin− (CD31−CD45−TER119−) MECs were selected by using a LS column adapter (Miltenyi Biotec, 130-042-401). Single-cell suspensions of MECs were suspended in 2% bovine serum albumin (BSA; Sigma) in phosphate-buffered saline (PBS, pH 7.4) and stained for surface expression of Cd24 (1:400, BD, 553262) and Cd29 (1:100, BioLegend, 102206) using optimized concentrations of fluorochrome-conjugated monoclonal antibodies (mAbs). Single-cell suspensions of MECs from MMTV-PyMT transgenic mice were stained with anti-F4/80 (1:100, BioLegend,

123110), anti-Cd11b (1:100, BioLegend, 101227), anti-Ly-6G/Ly-6C (Gr-1) (1:100, BioLegend, 108412), anti-Cd45 (1:100, 103108, BioLegend), Cd24 (1:400, BD, 553262), and Cd90 (1:400, BioLegend, 105307) fluorochrome-conjugated mAbs. F4/80[+]Cd11b[+]Gr[−] mouse mammary TAMs and Cd24[+]Cd90[+] mouse MECs were sorted by using a BD FACSAria[TM] cytometer (BD Biosciences)[60,61]. Flow cytometry data were collected using a BD FACSVerse[TM] and BD Accuri[TM] C6 plus cytometer (BD Biosciences) and analyzed using FlowJo software (Tree Star, Inc.).

Single-cell suspensions of human breast cancer MCF7 cells cultured in serum-free medium were stained with phycoerythrin (PE)-conjugated anti-CD44 (1:400, 338808) and fluorescein isothiocyanate-conjugated anti-CD24 (1:100, 311108) antibodies (Biolegend). CD44[Br]CD24[Di] and non-CD44[Br]CD24[Di] BCCs were sorted by using a BD FACSAria[TM] flow cytometer (BD Biosciences).

**Tissue microarray and immunohistochemistry.** Formalin-fixed, paraffin-embedded human breast cancer tissue and non-cancer tissue microarrays (BR1921c) were purchased from Alenabio (China). The human breast cancer microarray contained 80 cases of invasive ductal carcinoma, 80 cases of invasive lobular breast carcinoma, 24 cases of cancer adjacent normal breast tissue, and 7 cases of normal breast tissue.

The human tissue microarrays were deparaffinized in xylene, hydrated (100%, 95%, 80%, 70%, 0% of ethanol and water, 3 min each), and then antigen-retrieved in the presence of citrate buffer (10 mM sodium citrate, pH 6). Slides were permeabilized by 0.5% Triton X-100 for 30 min and incubated in 3% hydrogen peroxide for 15 min. After blocking with 3% BSA for 30 min at room temperature, slides were stained with the anti-TRIB3 (1:200, Abcam, ab137526), anti-FOXO1 (1:100, Abcam, ab52857), or anti-SOX2 (1:00, Abcam, ab171380) antibody overnight at 4 °C. After washing three times with PBS, slides were incubated with secondary antibodies for 60 min at room temperature. Following three times washes with PBS and staining with horseradish peroxidase (HRP)-conjugated streptavidin for 20 min at room temperature, the slides were developed for 1–10 min using DAB buffer and were rinsed in distilled water. Counterstaining was performed with hematoxylin for 10 s, followed by 10 min in distilled water. After dehydration (80%, 95%, and 100% ethanol followed by xylene), slides were mounted with Resinene. Images were obtained using an upright microscope (Olympus Microsystems).

**Immunofluorescence analysis and imaging.** The frozen primary or PDX breast cancer tissues were fixed with 4% paraformaldehyde, permeabilized in 0.5% Triton X-100, blocked with 3% BSA, and stained with an anti-TRIB3 (1:100, Abcam, ab137526), anti-CD68 (1:100, Abcam, ab955), anti-FOXO1 (1:100, Abcam, ab52857), or anti-SOX2 (1:100, Abcam, ab171380) primary antibody followed by Alexa Fluro 488 and/or Alexa Fluro594 secondary antibodies (1:200, Life Technologies). The cell nuclei were stained with 4′,6-diamidino-2-phenylindole. Fluorescence images were obtained by using a laser scanning confocal imaging system (Olympus Microsystems).

**Quantitative PCR.** Total RNA was extracted using Trizol (TransGen Biotech, China) according to the manufacturer's instructions. cDNA synthesis was performed with 0.5~1 μg of total RNA at 42 °C for 30 min by using TransScript II One-Step RT-PCR SuperMix (TransGen Biotech). mRNA levels were measured with gene-specific primers listed in Supplementary Table 4 by using the KAPA SYBR FAST qRT-PCR Kit (Kapa Biosystems, KK4601). The results were normalized to β-actin or GAPDH. The primers used of the indicated genes are shown in Supplementary Table 4.

**Chromatin immunoprecipitation.** EZ-Zyme chromatin preparation kits (17-375) and ChIP kits (17-371) were obtained from Merck Millipore (Billerica, MA, USA). Briefly, cells were cross-linked with 1% formaldehyde for 10 min at room temperature and redundant formaldehyde was inactivated by the addition of glycine[7]. The chromatin extracts containing DNA fragments were IP using anti-RNA Polymerase, normal mouse IgG, and FOXO1 or SOX2 antibody. The ChIP-enriched DNA was then decrosslinked and analyzed by real-time PCR. The primers used to amplify specific regions of the indicated genes are shown in Supplementary Table 4.

**Coimmunoprecipitation.** Cell pellets were lysed with coimmunoprecipitation buffer (25 mM Tri-cl (pH 7.4), 150 mM NaCl, 0.5% NP-40, 2.5 mM MgCl, 0.5 mM EDTA, 5% Glycerol), incubated with IP antibodies overnight at 4 °C with shaking, and then incubated with Protein A/G Plus-Agarose (Santa Cruz Biotechnology) for 2 h at 4 °C[15]. Interaction complexes were separated from the beads by boiling for 10 min and then subjected to SDS-PAGE and detected with immunoblotting.

**Mass spectrometry analysis.** Whole-cell lysates of MCF7 cells were IP with anti-FOXO1 antibody (CST, 14952) or anti-IgG1 isotype control antibody (CST, 5415) overnight at 4 °C, then incubated with Protein A/G Plus-Agarose (Santa Cruz Biotechnology) for 2 hr at 4 °C. Interaction complexes were separated from the

beads by heating at 98 °C for 10 minutes. Mass spectrometric data analysis was performed by Beijing Qinglian Biotech, Co., Ltd.

**Ubiquitination assay.** In vitro ubiquitination was performed according to the protocol provided by the Ubiquitination Kit (BML-UW9920, BioMol) from Enzo Life Sciences, Inc. Briefly, FOXO1-S256A-His and FOXO1-K463R-His were expressed in FreeStyle 293 cells and purified by a HisTrap FF column (GE) and ÄKTA pure protein purification system (GE)[62]. Purified recombinant Homo sapiens FOXO1 (TP300477) and NEDD4L (TP327866) proteins were purchased from Sino Biological, Inc. (China). Pyrophosphatase, inorganic from baker's yeast (Saccharomyces cerevisiae, 83205) were purchased from Sigma Aldrich. The assays were carried out at 37 °C in a 25 μl reaction mixture containing 20 U/ml inorganic pyrophosphatase (Sigma Aldrich), 1 mM dithiothreitol, 5 mM Mg-ATP, 100 nM E1, 2 μM indicated E2 (UBCH5b), 100 nM E3 (NEDD4L), 1 μM substrate protein (FOXO1), and 2.5 μM biotin-labeled ubiquitin. After incubation for 60 min, the reactions were IP with the anti-FOXO1 antibody for further analysis.

In vivo ubiquitination was carried out as described previously[63]. Briefly, HMLE or MCF7 cells were transfected with 6× His-ubiquitin and indicated vectors for 24 h. Cells were incubated with 20 μM MG132 for 6 h at 37 °C before harvest cells. One-tenth of whole cells were lysed with RIPA buffer containing protease inhibitors (Roche). Remaining cells were lysed in urea buffer (6 M urea, 0.1 M NaH₂PO₄, 0.1 M Tris-HCl, pH 8.0, 0.05% Tween-20, and 10 mM imidazole, pH 8.0) at room temperature for 1 h. The insoluble fraction was removed by centrifugation. Equal amounts of protein were incubated with Ni-NTA agarose beads (Qiagen) overnight at 4 °C. Beads were extensively washed four times with denaturing wash buffer (6 M urea, 0.1 M NaH₂PO₄, 0.1 M Tris-HCl, pH 8.0, 0.05% Tween-20, 20 mM imidazole, pH 8.0) and once with native wash buffer (0.1 M NaH₂PO₄, 0.1 M Tris-HCl, pH 8.0, 0.05% Tween-20, 20 mM imidazole, pH 8.0). Protein complexes were eluted in 500 mM imidazole for 20 min at room temperature. The ubiquitinated proteins were separated by SDS-PAGE for immunoblot analysis.

**Luciferase reporter assay.** HMLE or MCF7 cells were seeded 24 h before transfection. The FOXO1-GLuc (HPRM22487-PG04) and SOX2-GLuc (HPRM15202-PG04) vectors were obtained from GeneCopoeia, Inc. At 48-72 h post transfection, the cell culture medium was collected for GLuc and SEAP luminescent assays by using the Secrete-Pair Dual Luminescence Assay Kit according to the manufacturer's instruction (GeneCopoeia).

**Electrophoretic mobility shift assay.** Nuclear extracts of MCF7-SOX2-Myc cells were obtained by using NE-PER Nuclear and Cytoplasmic Extraction Reagents (Pierce). EMSAs were performed by using LightShift Chemiluminescent EMSA Kits (Pierce, 20148) as previously described[27]. Briefly, nuclear extracts were incubated with an anti-SOX2 antibody for supershift for 10 min at room temperaure (RT) in a reaction buffer. Then, biotin-labeled probes with or without unlabeled probes for the competitive reaction were added into the reaction system and incubated for 20 min at RT. Samples were separated in a 6% polyacrylamide gel and transferred onto a nylon membrane (Pierce), cross-linked by UV for 20 min, blocked with blocking buffer for 15 min, probed with streptavidin-HRP conjugates and incubated with the detection substrates. The probe sequences used for detection are shown in Supplementary Table 4.

**Homology modeling of the TRIB3 protein.** The amino-acid sequences of TRIB3 (NP_066981.2) were retrieved from the NCBI database and used as targets for homology modeling using Discovery Studio 2016 (BIOVIA). Create homology models modules were used to perform target-template sequence alignment after searching the putative X-ray template proteins in the NCBI server for generating the 3D models[64]. The structures of the top ten hits, including 5CEM, 5CEK, 4IXP, 4BL1, 4D2P, 4UMT, 4B6L, 2YZA, 2VN9, and 2WEL, were loaded and their sequences were aligned to the TRIB3 protein template sequences. The Build homology models module was then used to create the TRIB3 protein structure. The interactions between TRIB3 and AKT1 (PDB: 4GV1), peptide, and TRIB3 were predicted by the Dock Proteins (ZDOCK) module.

**Animal studies.** NOD-SCID mice (Vital River Lab Animal Technology, Beijing, China) and NPG (NOD-Prkdc[scid] Il2rg[null]) mice (Beijing Vitalstar Biotechnology) were housed in maximum barrier facilities, with individually ventilated cages, sterilized food, and water. FVB mice (Vital River Lab Animal Technology, China), MMTV-PyMT transgenic mice and MMTV-ErbB2 transgenic mice (Model Animal Research Center of Nanjing University, Jackson Laboratory) were maintained in the animal facility at the Institute of Materia Medica under specific-pathogen-free conditions. All mice were used at 6–8 weeks of age. Female mice were hosted for all breast cancer models. All experiments using animals were performed following protocols approved by the Animal Experimentation Ethics Committee of the Chinese Academy of Medical Sciences, and all procedures were conducted following with the guidelines of the Institutional Animal Care and Use Committees

of the Chinese Academy of Medical Sciences. All animal procedures were consistent with the ARRIVE guidelines[65].

Tumor cells were suspended in PBS with 50% Matrigel (354230, Corning) on ice before xenograft. The indicated numbers of MCF7 or MDA-MB-231 cells were transplanted into the 4th mammary fat pads of 6-week-old female NOD/SCID mice. Tumorigenesis was analyzed at 4 or 6 weeks after transplantation. The day before MCF7 cell transplantation, mice were subcutaneously implanted with 17β-estradiol control release pellets (SE-121, Innovative Research of America). The indicated numbers of primary mouse BCCs from MMTV-PyMT transgenic mice were injected into the 4th mammary fat pads of 6-week-old female FVB mice. PDX MECs were isolated from NPG mice by using the Tumor Dissociation Kit human (Miltenyi Biotec, 130-095-929) and injected into the 4th mammary fat pad of 6-week-old female NPG mice. One day after xenograft, the mice were intraperitoneally treated with 2 mg/kg Pep2–con or 2 mg/kg Pep2–Ae twice a week for 4 or 5 weeks. Tumorigenesis was analyzed at 4 or 5 weeks after transplantation. The last peptide treatments were provided one hour before primary BCCs isolation. The cell lysates of the isolated primary BCCs were IP with anti-AKT (1:100, CST, 2938) antibody or probed with the indicated antibodies for protein expression analysis by Western blotting.

**Human subjects.** Human breast cancer tissues were obtained from Anyang Cancer Hospital, Henan University of Science and Technology. Primary breast cancer fragments were mechanically minced before implantation[66]. The clinical features of the patients are listed in Supplementary Table 3. All protocols using human specimens were approved by the Institutional Review Board of the Chinese Academy of Medical Sciences and Peking Union Medical College. Informed consent was obtained from all subjects. The study conforms to the principles outlined in the Declaration of Helsinki.

**Generation of PDX animal models.** Fresh breast cancer tissues were spliced into small fragments (1–3 mm$^3$) in the medium. The tissue fragments were suspended in diluted Matrigel (Corning, 354248) 1:1 with PBS, and subcutaneously implanted into NPG mice (NOD-Prkdc$^{scid}$ Il2rg$^{null}$, Beijing Vitalstar Biotechnology). Early passages (1–5) of primary tumor tissues from these PDX models were mechanically minced and dissociated using gentleMACS$^{TM}$ Dissociator (Miltenyi Biotec) in accordance to the manufacturer's protocols[2].

**Mammosphere assays.** For mammosphere assays, PDX cells were seeded on 96-well ultralow attachment plates (Corning) at a density of 500 cells/well in StemXVivo Serum-Free Tumorsphere Media (R&D, CCM012) in the presence of the indicated peptides. Images were acquired by using a phase contrast microscope (Olympus Microsystems), and spheres were counted 5-7 days later. Mammosphere assays with MMTV-PyMT or MMTV-ErbB2 MECs were performed as described previously[67]. Single cells were plated in 24-well ultralow attachment plates (Corning) with sphere culture liquid medium. DMEM and Ham's F12 (50/50 Mix) media were used as the sphere culture liquid medium, and supplemented with B27 (Gibco), 20 ng/ml epidermal growth factor, 20 ng/ml basic fibroblast growth factor (Peprotech), and 4 mg/ml heparin. The spheres were counted 5–7 days later. MCF7 cells were cultured in 24-well ultralow attachment plates with liquid sphere culture medium for 5 days. The spheres were collected, and protein analysis was conducted after stimulation with 10 ng/ml IL-6 for 12 h, 50 mM glucose (HG) for 24 h, or 200 μM CoCl$_2$ for 24 h. Mammosphere assays with indicated cells were performed as previously described[68]. Resuspended cell solutions (10$^4$ cells/ml) were mixed with the same volume of Salmon fibrinogen (Sea Run Holdings, SEA-133), and the cell mixtures were seeded into 24-well plates that were pre-treated 5 μl of salmon thrombin (0.1 U/μl) (SEA-135, Sea Run Holdings). Thirty minutes after incubation at 37 °C in a cell culture incubator, 1 ml of complete medium was then added. The tumor spheres were counted at 4–7 days later.

Isolated tumor MECs from MMTV-PyMT transgenic mice were co-cultured with sorted mouse TAMs to conduct a mammosphere assay. The mouse mammary TAM cells were spin infected with GFP-adenovirus for 2 h at 1000 × g at 4 °C[69]. A total of 5000 MMTV-PyMT MECs were mixed with 20,000 TAMs and grown in low-adherence plates in sphere culture liquid medium[70]. Spheres were counted 5–7 days later. IL-6 expression in the co-cultured medium was analyzed by ELISA (LEGEND MAX™ Mouse IL-6 ELISA Kit, BioLegend, 431307) according to the manufacturer's instructions.

**Gene-set enrichment analysis.** Publicly available TCGA gene expression data of 1215 breast cancer samples were downloaded from the UCSC Xena. Of these cases, the upper-tenth (121, TRIB3 positively correlated) had the highest levels of TRIB3 expression, whereas the lower-tenth (121, TRIB3 negatively correlated) had the lowest levels of TRIB3 expression. Using the signal-to-noise measure in the GSEA, we ranked 20,530 genes by their association with the breast cancer groups (TRIB3 positively correlated vs. TRIB3 negatively correlated). Pre-ranked GSEA was performed using a dataset of the 380 ES/stem cell-associated genes as previously described[66,71].

**Gene expression analysis from the public database.** We conducted overall survival analysis of breast cancer patients from the TCGA-BRCA database by using the Kaplan–Meier plotter (http://kmplot.com)[20]. Patients were subsequently sub-grouped into the upper tertile (TRIB3$_H$), intermediate tertile and lower tertile (TRIB3$_{L+M}$) based on their relative expression of TRIB3 (Probe: 218145_at). For relapse analysis, we compiled four microarray datasets (GSE2603, GSE5327, GSE2034, and GSE12276) of 582 patients from the PubMed GEO database (Gene Expression Omnibus database, http://www.ncbi.nlm.nih.gov/gds). The normalized microarray datasets were center to the median of all probes[21,72]. Patients were subsequently sub-grouped in tertiles based on their relative expression of TRIB3 (Probe: 218145_at).

**Surface plasmon resonance analysis.** Peptide Ae, alanine mutants of Ae (M1-M13), Pep2–con and Pep2–Ae were synthesized by Guoping Pharmaceutical, Co. (China). The Surface plasmon resonance binding kinetics between TRIB3 and the indicated peptides were analyzed by using a BIAcore T200 system (GE Healthcare)[14]. Briefly, TRIB3-GST protein (Sino, China) was immobilized onto four individual flow cells on a CM5 sensor chip (GE Healthcare) through a standard coupling protocol. To measure the binding kinetics, the indicated peptide (2.5~0.009 μM in 2-fold serial dilutions) and a buffer blank for baseline subtraction were sequentially injected, with a regeneration step (glycine, pH 2.5) inserted between each cycle. The equilibrium dissociation constant (KD) was calculated according to BIA-evaluation software.

**RNA microarray assay.** MCF7-Con-siRNA and MCF7-TRIB3-siRNA1 cells were lysed with Trizol (TransGen Biotech, China). RNA microarrays were performed by KangChen Biotech, Inc. (China) using Gene Expression Array in compliance with MIAME guidelines. The GEO accession number is GSE116671.

**Statistical analysis.** The association of TRIB3 with treatment-free survival was assessed by Cox proportional hazards regression. Data are shown as the mean ± SEM. Student's t-test was used for two-group comparisons. Comparisons between three or more groups were analyzed by one-way analysis of variance followed by Duncan's test, unless otherwise indicated, using GraphPad Prism 6 or SPSS 17.0 (SPSS, Inc.). $P < 0.05$ was considered statistically significant.

## Data availability

All microarray data generated in this study have been deposited at the NCBI Gene Expression Omnibus with the accession code GSE116671. The KM plotter breast cancer dataset was obtained from http://kmplot.com/analysis. The mass spectrometry proteomics data have been deposited to the ProteomeXchange Consortium via the iProX partner repository with the dataset identifier PXD014817 (http://proteomecentral. proteomexchange.org/cgi/ GetDataset?ID = PXD014817). All other data supporting the findings of this study are available from the corresponding authors upon reasonable request. A Reporting Summary for this study is available as a Supplementary Information file. The uncropped blot figures, flow cytometry gating strategy, and original data underlying Figs. 1–8 and Supplementary Figs. 1–10 are provided as a Source Data file.

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

## Acknowledgements

We thank Professor Xiaoming Zhou (Suzhou University, China) for providing His 6-Ubiqutin vector. This work was supported by grants from the National Key R&D Program of China (2017YFA0205400), the National Natural Science Foundation of China (81773781, 81530093, 81874316, 81603129, 81622010, and 81770800), the Drug Innovation Major Project from the National Science and Technology (2018ZX09711001-003-001 and 2018ZX09711001-003-009), the Chinese Academy of Medical Sciences (CAMS) Central Public-interest Scientific Institution Basal Research Fund (2017PT31046 and 2018RC350004), the CAMS Innovation Found for Medical Sciences (2016-I2M-3-008, 2016-I2M-1-007, 2016-I2M-1-011, and 2016-I2M-4-001), and Beijing Outstanding Young Scientist Program (BJJWZYJH01201910023028).

## Author contribution

B.C. and Z.W.H. conceptualized and participated in the overall design, supervision, and coordination of the study. J.M.Y., W.S. and Z.H.W. designed and performed most of the experiments. X.L., F.H., K.L., X.X.L., X.W.Z., Y.Y.L. and F.W. participated in molecular and cellular biological experiments. J.J.Y., S.S.L., S.S., Z.N.Y., C.X.Z., X.Y.H. and P.P.L. performed the animal studies. B.C., Z.W.H., J.M.Y., Z.H.W. and B.H. wrote the manuscript. All authors read and approved the manuscript.

## Competing interests

The authors declare no competing interests.
