## [Peer Review File · Nature Communications]

Reviewers' comments:

Reviewer #1, Expertise: Foxo, cancer
(Remarks to the Author):

In this manuscript (NCOMMS-18-29005), Yu et al. present data that lead to an interesting model, in which high Trib3 expression results in impaired AKT activity, increased FOXO1 stability and activity, increased SOX2 expression and ultimately enhanced breast cancer stemness. FOXO1 and SOX2 are also implicated in an autoregulatory loop. Finally, the authors show that the whole pathway can be inhibited with a peptide that disrupt the first step, the TRIB3-AKT1 interaction. In general the manuscript is clearly written and the data presented are convincing. While potentially interesting, several improvements are required to support the claims made by the authors

General

TRIB3 regulation of AKT and AKT regulation of FOXOs is not novel, by now many E3 ligases are acclaimed to regulate FOXOs and the case put forward here is not particularly compelling. Finally SOX2 as a FOXO target could be relevant, but other equally well established regulators of 'stemness' e.g. myc are also regulated by FOXOs (negative c-myc) as well as positive (KLF4) so it represents to me a too narrow perspective to boil down all effects on SOX2 regulation.

Major points:

1. Throughout the manuscript the authors suggest that stress leads to TRIB3 up-regulation, which then contributes to breast cancer stemness. Experiments in which for example cells or spheroids are subjected to stress, leading to all the downstream mechanisms shown, are however not included. This would significantly improve the manuscript. Also what stress?
2. The authors selected E3 ligases that may degrade FOXO1 based on gene expression data in Trib3-silenced cells (p. 15). However, Tribbles proteins function as scaffold proteins and may regulate degradation in other ways (e.g. support de-ubiquitination). This should be addressed experimentally. In addition, on p.15 the authors state that "SKP2,...but not NEDD4L... was found to mediate FOXO1 degradation....", but Fig. S5D shows the opposite. Actually the whole ubiquitin part is confusing. Experimentally it is not done properly, flag pull downs contain a whole array of proteins (just do a mass spec on a flag pull down and you know how many) so the smear could represent any ubiquitinated protein possible and is certainly not specific enough. Also most other studies have shown that FOXOs are relatively stable proteins with half-life of around 12 hours, unless abnormal high PI3K expression or activity, so the cell lines used suggest hyperactive AKT to start with. This also suggests that the model drawn by the authors only accounts for those cells that have oncogenic AKT activity and not for cell lines/models without PI3K as a driver. Furthermore based upon the model AKT activation in general should result in FOXO
3. TRIB3 is a direct FOXO1 target gene in neurons (PMID 24212932), so it is possible that this creates an additional feedforward loop. FOXOs in general though transcriptionally regulate AKT regulators mostly activators. This hypothesis should be addressed experimentally, whether we are dealing with a feedforward or not as this also cautions interpretation of data.
4. The authors used Pep2-Ae and a control peptide to inhibit the TRIB3-AKT1 interaction. In this type of experiment, it is crucial to assess the specificity of the inhibition; ideally the control peptide should only differ minimally from Pep2-Ae in sequence, but maximally in function. The sequences of Pep2-Ae and Pep2-con are not given, which prevents proper evaluation. If Pep2-con differs substantially from Pep2-Ae, additional binding experiments (e.g Fig. 7) should be performed to identify a proper Pep2-con.
5. Several mistakes are made when referring to the different panels of Fig. S6 and S7 (p. 17). Also, the title of Fig. S7 seems incorrect, as only panel E concerns the Akt1-FOXO1 interaction.

Minor points:

1. p.4 Replace 'nucleous' with 'nucleus' in figure.
2. p.14: The authors describe the use of CQ, but Bafilomycin is indicated in Figure 5. Please correct.

3. p.16: Replace 'JUK' with 'JNK'.
4. p.18: Refer to Fig. S67E for endogenous co-IP.
5. p. 19: Clarify procedure when stating "created by homology models".
6. p. 37: Indicate number of experiments (n=...) for Fig. 3b, right panel and Fig. 3c, right panel.

Reviewer #2, Expertise: breast CSC (Remarks to the Author):

: In this manuscript the authors demonstrate that the pseudokinase TRIB3 is positively associated in breast cancer patients with tumor progression and breast cancer stemness. They define a molecular pathway by which TRIB3 interacts with AKT that, in turn, interacts with the FOXO1-AKT to suppress FOXO1 phosphorylation, ubiquitination and degradation by E3 ligases. FOXO1 in turn, promotes the expression of SOX2 which in turn, drives cancer stemness. Of importance, they generate a fused peptide which can interfere with the TRIB3-AKT1 interaction and demonstrate that this reduces cancer stemness and tumor initiation in mouse models. In general, the experiments are well done and thoroughly reported and define an important novel pathway by which stress induced TRIB3 is able to induce cancer stemness via the transcription factor SOX2. However, several additional experiments and issues need to be considered:

1. The authors need to better define the cellular heterogeneity of TRIB3 expression in human breast tumors and the relation of TRIB3 to cancer stemness. Figure 1B suggests that TRIB3 is expressed in a very high percentage of invasive breast cancer cells. However, according to previous reports the breast cancer stem cell population represents only a small subcomponent, i.e. 1-5% of cells within a breast tumor. How do the authors explain the discrepancy in TRIB3 expression at the cellular level with cancer stemness? Does this indicate that TRIB3 expression is necessary but not sufficient to drive SOX2 and stemness. A similar issue is raised by Figure 3D, which shows SOX2 expression in many more cancer cells than the expected in the stem cell population. The authors should perform co-staining for FOXO1 and SOX2 to determine whether these are co-expressed in individual cells.
2. Figure 1d & e show the probability of survival and relapse related to TRIB3 expression in a human breast cancer data set. Were these patients segregated for breast cancer subtype i.e. estrogen receptor positive, HER2 positive vs. triple negative? Does association of TRIB3 with prognosis relate across these molecular subtypes?
3. The authors contend that the CD24⁺/CD29^(low) population is the stem cell population in mice in the luminal type MMTV-PyMT breast cancer model. The authors need to directly demonstrate or provide a reference indicating that expression of these markers defines the stem cell phenotype in this model. In addition, other investigators have suggested that CD90^(thy1) represents the cancer stem cell phenotype in MMTV-PyMT tumors. Have the authors looked at this marker?
4. The authors show CD68⁺ tumor associated macrophages are located adjacent to areas of breast cancer (Figure F1h i). However, they provide little follow-up information for the role of these cells in TRIB3 induced stemness. They show that IL6 can regulate TRIB3 in vitro, and it is known that macrophages can secrete IL6 but this connection is not investigated.
5. In Figure 3a, the authors demonstrate that overexpression of TRIB3 in HMLE cells enhances protein expression of SOX2 in addition to NANOG and c-Myc. Silencing of TRIB3 in MCF7 cells reduces protein expression of SOX2 and NANOG. In MDA-MB-231 cells, silencing of TRIB3 expression attenuated the mRNA expression of SOX2 and KLF4 and protein expression of SOX2 and KLF4. There are several issues with these experiments that need to be clarified. Since SOX2 is selectively expressed in cancer stem cells; How do the authors explain their results when they look at expression in the total cell population rather than in cancer stem cells alone? Furthermore, these experiments suggest that in addition to SOX2, expression of other pluripotency factors including NANOG, c-Myc and KLF4 at the mRNA or protein level are also regulated by TRIB3. Although the authors have chosen to focus on investigating TRIB3/SOX2 in breast cancer stemness, they do not discuss the potential role of these other pluripotency factors. For example, have they determined whether knockdown of NANOG affects cancer stemness?
6. The authors demonstrate that TRIB3 knockdown reduces sphere formation and expression of CD44⁺CD24⁻ in vitro. Have the authors investigated whether SOX2 knockdown had similar effects

and whether there are any additional effects of TRIB3 knockdown in SOX2 knock downed cells, indicating potential additional pathways for cancer stem cell regulation.

7. In general, the experiments looking at the effects of TRIB3 on FOXO1 stability are well done. However, they demonstrate that TRIB3 enhanced the half-life of FOXO1 degradation from 1.5 to 7.2 hours in HMLE cells (Figure 5a). In contrast, silencing reduced the half-life FOXO1 from 3.1 to 0.7 hours. Why are the controls i.e. 1.5 hours in the overexpression and 3.1 hours in the degradation different? Does this indicate batch effects?

8. Figure 6d indicates that the AKT1 inhibitor MK2206 modulates the activity AKT1 and rescued FOXO1 and SOX2 expression. In order to demonstrate that this is directly due to the effects of this inhibitor on AKT1 and not due to off-target effects, it would strengthen the argument to use a genetic approach, i.e. AKT knockdown to mimic the the effects of the inhibitor.

9. The description of the Pep2-Ae peptide studies in a tumor xenograft are somewhat confusing, i.e. Figure 8e & 8f. The authors need to provide a clearer description of whether these experiments represent in vivo treatment in primary animals or secondary transplants. The gold standard test for cancer stem cell effects are to perform the treatment in primary animals and to assess tumor initiating capacity in secondary untreated animals. The authors need to more clearly describe how these experiments were performed.

10. Since TRIB3-AKT-FOXO1-SOX2 activates a positive feedback loop, do the authors have any information as to potential negative physiological signals which turn the pathway off?

Reviewer #3, Expertise: peptide design (Remarks to the Author):

The manuscript from Yu, et al. details the authors' efforts in teasing out a signaling pathway that is responsible for the stemness of breast cancer cells. Therein, the authors describe a very detailed study where they determined that the interaction between TRIB3 and Akt is crucial for breast cancer cell survival, and that it's inhibition portends a novel way to target TRIB-3 overexpressing breast cancers therapeutically.

The studies described therein are quite thorough and very well executed. It was great to see that the authors took this a step further by discovering a peptide sequence that could conceivably block the interaction in question. While the manuscript could use some work in terms of copy-editing, the scientific content is solid, thus, I would recommend it for publication in the journal pending stylistic corrections.

Reviewer #1, Expertise: Foxo, cancer (Remarks to the Author)

In this manuscript (NCOMMS-18-29005), Yu et al. present data that lead to an interesting model, in which high Trib3 expression results in impaired AKT activity, increased FOXO1 stability and activity, increased SOX2 expression and ultimately enhanced breast cancer stemness. FOXO1 and SOX2 are also implicated in an autoregulatory loop. Finally, the authors show that the whole pathway can be inhibited with a peptide that disrupt the first step, the TRIB3-AKT1 interaction. In general, the manuscript is clearly written and the data presented are convincing. While potentially interesting, several improvements are required to support the claims made by the authors

General

TRIB3 regulation of AKT and AKT regulation of FOXOs is not novel, by now many E3 ligases are acclaimed to regulate FOXOs and the case put forward here is not particularly compelling. Finally SOX2 as a FOXO target could be relevant, but other equally well established regulators of 'stemness' e.g. myc are also regulated by FOXOs (negative c-myc) as well as positive (KLF4) so it represents to me a too narrow perspective to boil down all effects on SOX2 regulation.

Re: We thank this reviewer for his encouraging words, professional criticisms, and constructive comments regarding our manuscript. As a scaffold protein, TRIB3 stands out for its diverse signaling pathways (Kung, Jennifer, et al., 2019, *Nat Rev Drug Discov*). Recent findings from our group have indicated that TRIB3 interacts with different proteins to support the tumor stemness in various cancers. In colon cancer, the TRIB3/ β -catenin/TCF4 heterodimer promotes colon cancer tumorigenesis and cancer stemness (Hua F et al., 2019, *Gastroenterology*). In acute promyelocytic leukemia (APL), TRIB3 interacts with oncoprotein PML-RAR α and enhances the stability of PML-RAR α , which promotes the self-renewal of APL-initiating cells (Li K et al., 2017, *Cancer Cell*). Moreover, TRIB3 interacts with autophagy protein p62 to promote cancer development in liver cancer, colon cancer, lung cancer and melanoma (Hua F et al., 2015, *Nat Commun*). These studies suggest that each type of cancer has its own key pluripotency factor and regulatory network to mediate or promote high TRIB3-driven tumorigenesis. These may confer an alternative angle and serve good examples for the cancer heterogeneities.

We agree with you that other pluripotency factors, beside SOX2, might also play multiple roles in supporting breast cancer stemness. Actually, in this study, we have found that FOXO1, a recently recognized pluripotency factor, coordinates with SOX2, to support breast cancer stemness. Additionally, following your suggestions, we examined tumor sphere formation to determine the relationship between TRIB3 and these pluripotency factors (Revised Supplementary Figure 3c). We found that TRIB3-promoted tumor sphere ability is dramatically reduced in SOX2- but not in c-MYC- or KLF4- silenced breast cancer cells. Moreover, SOX2 but not c-MYC or KLF4 is consistently upregulated by TRIB3 in several different types of BCCs (Fig. 3a-c, Supplementary Figure 3a-b). Thus, we think that SOX2 rather than other pluripotency factors, in coordination of FOXO1, plays a crucial role in supporting the stemness in breast cancer with high TRIB3 expression. However, we briefly discuss the potential roles of other pluripotent factors in the direction of breast cancer stemness in the revised manuscript.

Major points:

1. Throughout the manuscript the authors suggest that stress leads to TRIB3 up-regulation, which then contributes to breast cancer stemness. Experiments in which for example cells

or spheroids are subjected to stress, leading to all the downstream mechanisms shown, are however not included. This would significantly improve the manuscript. Also what stress?

Re: This is a good suggestion. Following your opinion, we have provided new evidences to show that cytokines, hypoxia, and high glucose can enhance TRIB3 expression (Revised Supplementary Figure 1d). In the MCF7 cells under spheroids culture conditions, 12-hr IL-6 stimulation, 24-hr CoCl₂-induced hypoxia, and 24-hr high-glucose treatment enhance phosphorylation of FOXO1 and the abundances of FOXO1 and SOX2 (Revised Supplementary Figure 7f). These evidences support our findings that TRIB3 links stressors to enhance breast cancer stemness by activating the TRIB3-AKT-FOXO1-SOX2 axis in high TRIB3 expressed breast cancer.

2. The authors selected E3 ligases that may degrade FOXO1 based on gene expression data in Trib3-silenced cells (p. 15). However, Tribbles proteins function as scaffold proteins and may regulate degradation in other ways (e.g. support de-ubiquitination). This should be addressed experimentally. In addition, on p.15 the authors state that “SKP2,...but not NEDD4L... was found to mediate FOXO1 degradation...”, but Fig. S5D shows the opposite. Actually the whole ubiquitin part is confusing. Experimentally it is not done properly, flag pull downs contain a whole array of proteins (just do a mass spec on a flag pull down and you know how many) so the smear could represent any ubiquitinated protein possible and is certainly not specific enough. Also most other studies have shown that FOXOs are relatively stable proteins with half-life of around 12 hours, unless abnormal high PI3K expression or activity, so the cell lines used suggest hyperactive AKT to start with. This also suggests that the model dawn by the authors only accounts for those cells that have oncogenic AKT activity and not for cell lines/models without PI3K as a driver. Furthermore based upon the model AKT activation in general should result in FOXO.

Re: We agree with your point that TRIB3 may interact with other ubiquitin-editing proteins. Following your suggestion, we conducted experiments to integrate the de-ubiquitination pathway for the regulation of FOXO1 degradation. By mass spectrometry analysis, USP5, USP14, and USP32 were found to act as FOXO1-interacting deubiquitination proteins in MCF7 cells. However, silencing USP5, USP14 or USP32 did not interfere with FOXO1 expression (Revised Supplementary Figure 5d). We then moved on and focused on E3 ligases.

Sorry about the confusion about E3 ligases of FOXO1. We found that both SKP2 and NEDD4L mediated FOXO1 degradation. However, SKP2, but not NEDD4L, was found to mediate FOXO1 degradation by S256 phosphorylation (Supplementary Fig. 5e). TRIB3 directly reduces the expression of NEDD4L, and NEDD4L degraded FOXO1 by K463-FOXO1 ubiquitination (Supplementary Fig. 6e). In response to the reviewer's questions regarding ubiquitin assays, we have performed additional experiments to confirm our findings. We immunoprecipitated ubiquitin and then detected FOXO1 protein by anti-Myc or anti-Flag antibodies (Revised Supplementary Fig. 5f and Revised Supplementary Fig. 6e). The *in vitro* ubiquitination assay has also been conducted with the purified proteins (Revised Supplementary Fig. 5g, Revised Supplementary Fig. 6f). With these results, we confirmed that both SKP2 and NEDD4L mediated FOXO1 degradation in BCCs. We proposed that FOXO1 is degraded by SKP2 through FOXO1-S256 phosphorylation and NEDD4L through FOXO1-K463 ubiquitination in breast cancer. To prevent confusion, we rephrased these sentences in the current manuscript as follows:

“SKP2 (Supplementary Fig. 5e-f, left), but not NEDD4L (Supplementary Fig. 5e-f, right), was found to mediate FOXO1 degradation by S256 phosphorylation.”

We noticed a relatively longer half-life of FOXO1 degradation reported in literature. For example, Matsuzaki et al. (2003, *PNAS*) found that the half-life of FOXO1 degradation is 12-hr in HEPG2 cells under sera deprivation. After insulin activation of the PI3K-AKT signaling pathway, the half-life of FOXO1 degradation is approximately 6-hr. We thus designed a 24-hr time course for detecting FOXO1 degradation in MCF7 cells in the presence of protein synthesis inhibitor CHX. However, we found that the half-life of FOXO1 was much shorter than these in literature. We think that though sera deprivation can make cells enter a quiescent state, there should still have some degree of basal protein synthesis in cells under these conditions. However, in our study, we measured the half-life of FOXO1 without de novo protein synthesis in the presence of protein synthesis inhibitor CHX. This may be a major reason for the discrepancies between literature and our data. To verify this, we have repeated these experiments several times. The results showed the half-life of FOXO1 is about 3-4 hr with CHX treatment in BCCs (Revised Figure 5a-c). Similar findings show that the half-life of FOXO1 is approximately 4-hr in 293T cells (Du et al., 2014, *J Immunol*) and HCT116 colon cancer cells (Chae et al., 2019, *Nucleic Acids Res*) with CHX treatment. To obtain an accurate and consistent half-life for FOXO1, we adjusted the time-course range from 24-hr intervals, 12-hr intervals until 8-hr intervals. In the presence of CHX, the half-life of FOXO1 in HepG2 cell was about 3-hr (Fig1. B). The half-life of FOXO1 was 3-hr in HMLE cells, 6-hr in HMLE-TRIB3 cells (Revised Figure 5a), 3-hr in MCF7 cells, 0.7-hr in MCF7-TRIB3 shRNA cells (Revised Figure 5b). The proteasome inhibitor MG132 prolonged the half-life of FOXO1 to 6-hr in MCF7 cells (Revised Figure 5c).

Figure 1. The basal level of AKT activity and FOXO1 degradation in HEPG2, HMLE, and MCF7 cells. (A) The basal level of AKT activity in HEPG2 cells, HMLE cells, and MCF7 cells. (B) The FOXO1 degradation under CHX treatment in HEPG2 cells. (C) The FOXO1 degradation with PI3K inhibitor LY294002 in MCF7 cells.

The basal AKT activity of HMLE cells and MCF7 cells was lower than that of HepG2 cells, indicating that these BCCs do not in the hyperactive AKT status (Fig1. A). As your assumption, PI3K inhibitor LY294002 increased the half-life of FOXO1 to 4.4-hr in MCF7 cells, which have oncogenic AKT activity (Fig1. C). However, TRIB3 showed a similar phenomenon in that it interacted with AKT and promoted FOXO1 nuclear retention in HMLE-TRIB3 cells, which is normal mammary epithelial cells with low AKT activity. It is very interesting to know the impact of PI3K on this pathway. Considering the complexity of AKT phosphorylation by PI3K-dependent or PI3K-independent pathways (Mahajan et al. 2012, *J Cell Physiol*) and content limitation, we emphasize the impact of TRIB3/AKT interaction for breast cancer stemness in the current manuscript, and briefly discussed the potential role of PI3K, which is upstream of AKT.

3. TRIB3 is a direct FOXO1 target gene in neurons (PMID 24212932), so it is possible that this creates an additional feedforward loop. FOXOs in general though transcriptionally regulate AKT regulators mostly activators. This hypothesis should be addressed

experimentally, whether we are dealing with a feedforward or not as this also cautions interpretation of data.

Re: Following your suggestion, we have added new experiments to examine if a similar feedforward loop exists in BCCs. We found that overexpression of FOXO1 enhances TRIB3 expression in HMLE cells and MCF7 cells (Revised Supplementary Fig. 7g). However, silencing FOXO1 could not completely suppress IL-6, hypoxia or glucose-deprivation induced *TRIB3* transcriptional expression (Revised Supplementary Fig. 7h). These results indicate that FOXO1 contributes to *TRIB3* transcription under stress conditions, which forms a positive feedback regulation for the TRIB3-FOXO1-SOX2 axis in supporting breast cancer stemness.

Overexpression of FOXO1 was accompanied by reduced AKT activities (Revised Supplementary Fig. 7g). However, some AKT activators genes (Chia-Chen et al., 2010, *Developmental Cell*) such as *SESN3*, *RICTOR*, *TSC2*, *FGFs (1,4,8)*, *FGFRs (1,4)*, et al. are indeed up-regulated in MCF7 *CTRL-shRNA* cells, which are highly expressed FOXO1. These might be a potential negative physical signal for TRIB3-FOXO1-SOX2 axis. However, we did not further examine the regulatory role of FOXO1 on AKT activators because of space limitation. A project involved in the study of the expression profiles of FOXO1 in the subtypes of breast cancer is conducting in our group, which will deal with this interesting matter.

4. The authors used Pep2-Ae and a control peptide to inhibit the TRIB3-AKT1 interaction. In this type of experiment, it is crucial to assess the specificity of the inhibition; ideally the control peptide should only differ minimally from Pep2-Ae in sequence, but maximally in function. The sequences of Pep2-Ae and Pep2-con are not given, which prevents proper evaluation. If Pep2-con differs substantially from Pep2-Ae, additional binding experiments (e.g Fig. 7) should be performed to identify a proper Pep2-con.

Re: We thank you for your pointing out this missed information in our previous manuscript. Now, the sequences of Pep2-Ae and Pep2-Con and screening strategy have been provided in the revised methods section, revised figure 7 and revised supplementary figure 9 in the revised manuscript. Briefly, we used the alanine scan to identify the critical amino acid for peptide Ae. We found 1V(M1), 5L(M5), 6T(M6), 11L(M11), 12Q(M12) and 13N(M13) are the key regions for Ae/TRIB3 interaction. We then chose AAHTAAENRVAAA as the control peptide. This control peptide showed no interaction with TRIB3. In response to the reviewer's suggestion, we incorporated the results for the specificity of Pep2-Ae. As shown in revised supplementary figure 9, Pep2-Ae showed no effects on TRIB3-interacting proteins (SMAD3, P62, and β -catenin) or AKT1-interacting proteins (PDK1, GSK3 β , and P85).

5. Several mistakes are made when referring to the different panels of Fig. S6 and S7 (p. 17). Also, the title of Fig. S7 seems incorrect, as only panel E concerns the Akt1-FOXO1 interaction.

Re: We thank the reviewer for point out these mistakes. We corrected all mistakes regarding Fig. S6 (Revised Supplementary Fig. 8) and S7 in the manuscript. We also revised the title of Fig. S7 as "AKT, but not other pathways, mediates the TRIB3-FOXO1-SOX2 axis".

Minor points:

1. p.4 Replace 'nucleous' with 'nucleus' in figure.

Re: We corrected 'nucleous' with 'nucleus' in Fig. 8. Thanks.

2. p.14: The authors describe the use of CQ, but Bafilomycin is indicated in Figure 5. Please correct.

Re: We initially used CQ as an autophagy inhibitor to perform the pilot study. We chose bafilomycin as the autophagy inhibitor to conduct the current study. We have corrected this mistake in the revised manuscript to indicate bafilomycin (BAF) as the selected autophagy inhibitor.

3. p.16: Replace 'JUK' with 'JNK'.

Re: We corrected all 'JUK' with 'JNK' on p.16 and in the figure legends.

4. p.18: Refer to Fig. S67E for endogenous co-IP.

Re: Thank you for your suggestion. We add the following description on p. 18: "TRIB3 did not endogenously interact with FOXO1 in MCF7 cells (Supplementary Fig. 7i)." We also revised the figure legends to indicate this endogenous co-IP results.

5. p. 19: Clarify procedure when stating "created by homology models".

Re: We added a paragraph in the method section to describe the details of the homology model creation by Discovery Studio.

6. p. 37: Indicate number of experiments (n=...) for Fig. 3b, right panel and Fig. 3c, right panel.

Re: All statistical analyses were done by three independent assays. We indicated the number of experiments in the current figure legends.

Reviewer #2, Expertise: breast CSC (Remarks to the Author):

In this manuscript the authors demonstrate that the pseudokinase TRIB3 is positively associated in breast cancer patients with tumor progression and breast cancer stemness. They define a molecular pathway by which TRIB3 interacts with AKT that, in turn, interacts with the FOXO1-AKT to suppress FOXO1 phosphorylation, ubiquitination and degradation by e3 ligases. FOXO1 in turn, promotes the expression of SOX2 which in turn, drives cancer stemness. Of importance, they generate a fused peptide which can interfere with the TRIB3-AKT1 interaction and demonstrate that this reduces cancer stemness and tumor initiation in mouse models. In general, the experiments are well done and thoroughly reported and define an important novel pathway by which stress induced TRIB3 is able induce cancer stemness via the transcription factor SOX2. However, several additional experiments and issues need to be considered:

1. The authors need better define the cellular heterogeneity of TRIB3 expression in human breast tumors and the relation of TRIB3 to cancer stemness. Figure 1B suggests that TRIB3 is expressed in a very high percentage of invasive breast cancer cells. However, according to previous reports the breast cancer stem cell population represents only a small subcomponent, i.e. 1-5% of cells within a breast tumor. How do the authors explain the discrepancy in TRIB3 expression at the cellular level with cancer stemness? (Q2) Does this indicate that TRIB3 expression is necessary but not sufficient to drive SOX2 and stemness. A similar issue is raised by Figure 3D, which shows SOX2 expression in many more cancer cells than the expected in the stem cell population. (Q3) The authors should perform co-staining for FOXO1 and SOX2 to determine whether these are Co-expressed in individual cells.

Re: The major findings in this study were that the TRIB3-elevated breast cancer cells (BCCs) show enhanced stemness ability due to the coordination of elevated FOXO1 and SOX2 in these cells. Indeed, elevated TRIB3 expression was found in a very high percentage of invasive BCCs. Even with these results, we could not claim that TRIB3 is a specific stem cell marker. However, the results of *in vitro* tumor sphere and *in vivo* tumor formation assays indicate that TRIB3 triggers a signaling axis activation that supports the breast cancer stemness. We agree with the point that breast cancer stem cells need multiple parameters to be defined. The expression of a single stemness-contributing protein might have a relatively broad range. For example, ALDH1+ cells range from 4% to 19% in invasive breast cancer (Ricardo S et al., 2011, *J Clin Pathol*), and SOX2+ cells range from very rare (<2%) pattern to common expressed pattern (>65%) in breast cancer (Leis O et al., 2012, *Oncogene*). Importantly, recent work indicates that the cancer stem cells and common cancer cells can be free transited each other, depending on the tumor microenvironment, which is named as so-called stem cell plasticity. Our major findings that the stressors existing in the tumor microenvironment enhances TRIB3 expression, which activates the AKT-FOXO1-SOX2 axis to make the breast cancer cells with high TRIB3 expression transiting to the breast cancer cells with the stemness. We have briefly discussed these in the revised manuscript.

Thus, we think that TRIB3 is sufficient but not necessary to drive SOX2 and stemness. The BCCs probably have many other ways to maintain the stemness. Following your suggestion, we expanded SOX2 IHC staining for human breast cancer tissue array. The SOX2 positive range is broad in BCCs. This observation is similar to reports from other groups (Leis et al., 2012, *Oncogene*). We choose the representative IHC images with median levels of SOX2. Overall, the SOX2 positive expression is below 5% in above 90% of breast cancer patients.

We have also examined the double-positive BCCs for FOXO1 and SOX2 by IHC staining from the tissue microarray (Revised Supplementary Fig. 4f). By using confocal microscopy, we identified the SOX2⁺FOXO1⁺ cells in the patient-derived engrafted tumors (Revised Supplementary Fig. 4g). These data indicate that elevated expressions of FOXO1 and SOX2 in individual BCCs and these data have been added in the revised manuscript.

2. Figure 1d & e show the probability of survival and relapse related to TRIB3 expression in a human breast cancer data set. Were these patients segregated for breast cancer subtype i.e. estrogen receptor positive, HER2 positive vs. triple negative? Does association of TRIB3 with prognosis relate across these molecular subtypes?

Re: Yes, you are right. TRIB3 associates with overall survival (TCGA dataset) and relapse (Joan's dataset) in ER positive, HER2 positive and triple negative breast cancer patients. However, high TRIB3 expression per se has independent prognostic values. We have added overall survival analysis and relapse analysis based on breast cancer molecular subtypes (Revised Supplementary Fig 1b-c, Revised Supplementary Table 1 and 2). The overall survival analysis is based on Kaplan-Meier plotter platform (<http://kmplot.com>). To determine the independent prognosis of TRIB3, multivariable Cox regression analysis of relapse was performed. The data show that TRIB3 expression has independent prognostic values in overall relapse (HR=3.0), lung relapse (HR=2.7) and bone relapse (HR=2.8).

3. The authors contend that the CD24⁺/CD29^(low) population is the stem cell population in mice in the luminal type MMTV-PyMT breast cancer model. The authors need to directly demonstrate or provide a reference indicating that expression of these markers defines the stem cell phenotype in this model. In addition, other investigators have suggested that CD90^(thy1) represents the cancer stem cell phenotype in MMTV-PyMt tumors. Have the authors looked at this marker?

Re: Thank you for your constructive suggestion. We should only claim that the CD24⁺/CD29^(low) population includes tumor-initiating cells but does not represent tumor stem cells. Following your suggestion, we isolated the CD24⁺/CD90⁺/Lin⁻ subpopulation in MMTV-PyMT mice. We found that the higher *Trib3* expression in the CD24⁺/CD90⁺ subpopulation compares with that in non-CD24⁺/CD90⁺ subpopulation (Revised Supplementary Fig 1i). These results support the hypothesis that tumor-initiating cells have higher TRIB3 expression.

4. The authors show CD68⁺ tumor associated macrophages are located adjacent to areas of breast cancer (Figure F1h i). However, they provide little follow-up information for the role of these cells in TRIB3 induced stemness. They show that IL6 can regulate TRIB3 in vitro, and its known that macrophages can secretion Il6 but this connection is not investigated.

Re: We isolated MMTV-PyMT-derived tumor-associated macrophages (TAMs, F4/80⁺CD11b⁺Gr1⁻). These macrophage cells were mixed 4:1 with MMTV-PyMT-derived BCCs (Revised Supplementary Fig 1k, left panel). Per the reviewer's suggestion, IL-6 is known as one of the major cytokines released from TAM (Roy et al., 2014, *Immunity*). Silencing *Trib3* in breast cancer does not reduce IL-6 level (Revised Supplementary Fig 1k, middle panel), suggesting that TAMs are the major source for IL-6 in this co-culture system. In stem cell culture media, TAMs significantly enhance breast cancer stemness (Revised Supplementary Fig 1k, Right panel). IL-6 can enhance TRIB3 expression in breast cancer. We confirmed that blocking IL-6 significantly reduced tumor sphere. Silencing *Trib3* significantly reduced tumor sphere formation. These TAM niche-related phenomena provide

us clues to continue studying the impact of TRIB3 for breast cancer stemness. We incorporated these data and briefly discussed in the current manuscript.

5. In Figure 3a, the authors demonstrate that overexpression of TRIB3 in HMLE cells enhances protein expression of SOX2 in addition to NANOG and c-Myc. Silencing of TRIB3 in MCF7 cells reduces protein expression of SOX2 and NANOG. In MDA-MB-231 cells, silencing of TRIB3 expression attenuated the mRNA expression of SOX2 and KLF4 and protein expression of SOX2 and KLF4. There are several issues with these experiments that need to be clarified. Since SOX2 is selectively expressed in cancer stem cells; How do the authors explain their results when they look at expression in the total cell population rather than in cancer stem cells alone? Furthermore, these experiments suggest that in addition to SOX2, expression of other pluripotency factors including NANOG, c-Myc and KLF4 at the mRNA or protein level are also regulated by TRIB3. Although the authors have chosen to focus on investigating TRIB3/SOX2 in breast cancer stemness, they do not discuss the potential role of these other pluripotency factors. For example, have they determined whether knockdown of NANOG effects cancer stemness?

Re: We thank the reviewer for the suggestion. We have the same concern in terms of conducting stemness-related experiments. It would be ideal to check protein expression in the cancer stem cell population. However, no such cancer stem cell isolation method fits for all different types of breast epithelial cells, including human mammary epithelial cells (HMLE) and BCCs (MCF7, MDA-MB-231, etc.). In Figure 3a-3c, we analyzed the RNA and protein expression in isolated tumorspheres that were cultured under 3D mammary stem cell culture conditions but not a 2D cell culture system. These 3D culture conditions might amplify the effects of cancer stem cells. Moreover, recent studies suggested that the signals in the cancer stem cell niche can convert non-cancer stem cells to cancer stem cells (2017 Nat Medicine). These evidences suggested the importance of stemness activation loop other than stem cells themselves. We did not clarify the culture conditions in the previous manuscript. We now provide detailed information in the current results and method section and briefly discuss this inadequate analysis in the current manuscript.

We agree with your point that we should at least discuss the potential roles of other pluripotency factors including NANOG, c-Myc, and KLF4 in TRIB3-supported breast cancer stemness. Following your suggestion, we evaluated the capacity of tumor spheroids formation by silencing each of these pluripotency factors SOX2, NANOG, OCT4, KLF4 and c-Myc respectively in HMLE, HMLE-TRIB3, MCF7 and MCF7-TRIB3 shRNA cells (Revised Supplementary Figure 3c). We found that only silencing SOX2 but not other pluripotency factors significantly suppress tumor spheres formation in HMLE cells. In HMLE-TRIB3 cells, silencing any one of SOX2, NANOG, c-Myc can reduce tumor spheres formation. However, only silencing SOX2 can completely block the TRIB3-enhanced tumor spheres formation. In MCF7 cells, silencing any one of SOX2, KLF4, c-Myc, Nanog can significantly reduce tumor spheres. Silencing TRIB3 has additional effect in NANOG-shRNA cells, OCT4-shRNA cells, KLF4-shRNA cells, and c-Myc-shRNA cells, but not in SOX2-shRNA cells. These results indicate that SOX2 plays the most essential role in breast cancer stemness. Thus, TRIB3 supports cancer stemness mainly through enhancing SOX2 level. We briefly discuss these data and other group findings in the revised manuscript.

6. The authors demonstrate that TRIB3 knockdown reduces sphere formation and expression of CD44⁺24⁻ in vitro. Have the authors investigated whether SOX2 knockdown had similar effects and whether there are any additional effects of TRIB3 knockdown in SOX2 knock downed cells, indicating potential additional pathways for cancer stem cell regulation.

Re: We agree with your point. Following your suggestion, we examined the stemness in SOX2 knockdown cells (Revised Supplementary Fig 3f-g). We found that silencing SOX2 reduced CD44⁺CD24⁻ subpopulation from the original 3.1% to 0.2% and silencing TRIB3 reduces this subpopulation to 0.3%. This subpopulation was kept at 0.2% by silencing both SOX2 and TRIB3. Moreover, we examined tumor spheres formation using these cells (Revised Supplementary Fig3g). Silencing SOX2 reduces the number of tumor sphere to 1/3 that of control MCF7 cells. Silencing SOX2 and TRIB3 did not further reduce the capacity to form tumor sphere. These results indicate that TRIB3 reduces breast cancer stemness mainly through SOX2 regulation. However, due to the low percentage of CD44⁺CD24⁻ and low capacity of tumor sphere formation after silencing either TRIB3 or SOX2, it is still possible that there are additional pathways to regulate breast cancer stemness. We have discussed the possibility of these additional effects in the revised manuscript.

7. In general, the experiments looking at the effects of TRIB3 on FOXO1 stability are well done. However, they demonstrate that TRIB3 enhanced the half-life of FOXO1 degradation from 1.5 to 7.2 hours in HMLE cells (Figure 5a). In contrast, silencing reduced the half-life FOXO1 from 3.1 to 0.7 hours. Why are the controls i.e. 1.5 hours in the overexpression and 3.1 hours in the degradation different? Does this indicate batch effects?

Re: We thank this reviewer for pointing out this mistake. The discrepancy in the half-life of FOXO1 degradation might be resulted from the too wide ranges of time-course for these experiments. Following your suggestion, we adjusted the time points from 0, 2, 4, 8, and 12 h to 0, 0.5, 1, 2, 4, and 8 h, and this design has consistently been used for the whole project. New experiments show that the half-life of FOXO1 is approximately 3 hours for HMLE cells and MCF7 cells (Revised Fig. 5a, 5b). We repeated these assays for three times to avoid the batch effects.

8. Figure 6d indicates that the AKT1 inhibitor MK2206 modulates the activity AKT1 and rescued FOXO1 and SOX2 expression. In order to demonstrate that this is directly due to the effects of this inhibitor on AKT1 and not due to off-target effects, it would strengthen the argument to use a genetic approach, i.e. AKT knockdown to mimic the effects of the inhibitor.

Re: We agree with your point. We used a genetic approach to silence AKT1 (Revised Supplementary Fig7e). Silencing AKT1 rescued FOXO1 and SOX2 expression in MCF7-TRIB3-shRNA cells. This result plus the data of AKT1 inhibitor show that this is not an off-target effect.

9. The description of the Pep2-Ae peptide studies in a tumor xenograft are somewhat confusing, i.e. Figure 8e & 8f. The authors need to provide a clearer description of whether these experiments represent in vivo treatment in primary animals or secondary transplants. The gold standard test for cancer stem cell effects are to perform the treatment in primary animals and to assess tumor initiating capacity in secondary untreated animals. The authors need to more clearly describe how these experiments were performed.

Re: We agree with the point that secondary transplantation is the gold standard test for cancer stem cell effects. The Pep2-Ae peptide studies were performed in secondary untreated animals. We feel sorry not to provide a clear description in the previous methods section. We now amended the method description and revised figures. We conducted peptide treatment in primary animals (Revised Supplementary Fig. 10g, h) and secondary transplantation by using the isolated BCCs from treated animals to evaluate cancer initiation capacity (Figure 8e, 8f).

The peptide treatment show no effect for high dose primary tumor cell xenograft (MMTV-PyMT MECs xenograft, 5e4 cells/mouse; PDX xenograft, 1e4 cells/mouse), and a moderate effect for low-dose primary tumor cell xenograft (data not shown). To avoid confusion, we presented the results of high dose primary tumor cell xenograft, in which we analyzed protein expression (Figure 8f), protein interaction (Figure 8e), and isolated tumor cells for the secondary transplantation (Figure 8d).

10. Since TRIB3-AKT-FOXO1-SOX2 activates a positive feedback loop, do the authors have any information as to potential negative physiological signals which turn the pathway off?

Re: This is a really interesting point! To find out the negative physiological signals for this TRIB3-AKT-FOXO1-SOX2 loop, we should have a deep understanding of this stress sensor protein TRIB3 and other components. Unfortunately, the current study did not provide evidence to investigate these negative physiological signals. This may guide us to continue studying the potential impacts of this loop. However, we can make a “best-guess” attemptation as following:

First, as a stress response sensor, we expect that the removal of stress conditions can reduce TRIB3 expression, and then turn down this loop activation in TRIB3 highly-expressed breast cancer. In the niche microenvironments, this loop contributes to breast cancer stemness and maintains the low-proliferating stem-cell-like characteristics. However, certain AKT inducers might overcome the effects of TRIB3/AKT interaction, especially if cancer cells leave the TRIB3-enriched microenvironments. Second, FOXO1 activation could promote transcription of AKT activator genes (Ni et al., 2007, *PNAS*). Although FOXO1 nuclear retention was accompanied with reduced AKT activities in our study, some AKT activators genes (Chia-Chen et al, 2010, *Developmental Cell*) such as *SESN3*, *RICTOR*, *TSC2*, *FGF* ligands (*FGF1,4*, and *8*), *FGF* receptors (*FGFR1*, and *4*) et al. are indeed up-regulated in BCCs. These might be one potential negative physical signal. Third, AKT signaling has dual roles in regulating SOX2 levels by enhancing SOX2 protein stability and inhibiting SOX2 transcription in ESC, especially when SOX2 levels begin to rise above optimal levels. Jeong et al. (2010, *Stem Cells*) found that AKT directly phosphorylates SOX2, which increases SOX2 stability. However, signaling through AKT also inhibits SOX2 transcription (Ormsbee et al., 2013, *Plos One*). In ESCs, AKT inhibition actually enhances SOX2 transcription, compensate for the instability of SOX2 protein, and eventually increase SOX2 expression (Ormsbee et al., 2013, *Plos One*). As a result, it is possible that AKT inhibition might provide negative physiological signals by enhancing the vulnerability of SOX2 to degradation. We have briefly discussed the potential feedback regulations in the Discussion section in the revised MS.

Reviewer #3, Expertise: peptide design (Remarks to the Author):

The manuscript from Yu, et al. details the authors' efforts in teasing out a signaling pathway that is responsible for the stemness of breast cancer cells. Therein, the authors describe a very detailed study where they determined that the interaction between TRIB3 and Akt is crucial for breast cancer cell survival, and that its inhibition portends a novel way to target TRIB-3 overexpressing breast cancers therapeutically.

The studies described therein are quite thorough and very well executed. It was great to see that the authors took this a step further by discovering a peptide sequence that could conceivably block the interaction in question. While the manuscript could use some work in terms of copy-editing, the scientific content is solid, thus, I would recommend it for publication in the journal pending stylistic corrections.

Re: We thank this reviewer for his encouraging comments. Indeed, we are using the staple peptide technology to improve its druggability, etc. enhancing the stability and *in vivo* half-life of the Pep2-Ae peptide. We hope that the modified Pep2-Ae being a lead compound and yielding a better therapeutic efficacy against breast cancer.

Reviewers' comments:

Reviewer #1 (Remarks to the Author):

The revised manuscript by Yu et al. addressed my comments. The strategy chosen is to drown the reader (me) with figure after figure. This does not improve reading nor does it help the authors to provide a clear and compelling case for their major claim(s). As written in their rebuttal TRIB3 has been described to be involved in any possible mechanism depending on the tumor type (see quote from rebuttal below). My interpretation hereof is that apparently any claim can be made regarding TRIB3 each claim is valid and thus what do we learn ??? As mentioned initially in my first review, the essence of most of what is described here (TRIB3 acts on AKT, AKT regulates FOXO, FOXO is regulated through PTMs and UBI, FOXOs in stemness etc. is already known. so the advance presented here to me is minimal.

minor point: the Ubiquitin assay is still technically invalid. These should be performed using strong denaturing conditions (i.e. ip in Urea) in order to minimise the chance that one looks at UBI associated proteins rather than ubiquitinated proteins.

"Recent findings from our group have indicated that TRIB3 interacts with different proteins to support the tumor stemness in various cancers. In colon cancer, the TRIB3/ β -catenin/TCF4 heterodimer promotes colon cancer tumorigenesis and cancer stemness (Hua F et al., 2019, Gastroenterology). In acute promyelocytic leukemia (APL), TRIB3 interacts with oncoprotein PML-RAR α and enhances the stability of PML-RAR α , which promotes the self-renewal of APL-initiating cells (Li K et al., 2017, Cancer Cell). Moreover, TRIB3 interacts with autophagy protein p62 to promote cancer development in liver cancer, colon cancer, lung cancer and melanoma (Hua F et al., 2015, Nat Commun). These studies suggest that each type of cancer has its own key pluripotency factor and regulatory network to mediate or promote high TRIB3-driven tumorigenesis. If so then any claim on TRIB3 function "

Reviewer #2 (Remarks to the Author):

The authors have satisfactorily addressed my concerns including the inclusion of new data which significantly improves the work.
Some of the English requires editing.

Reviewer #3 (Remarks to the Author):

The authors have addressed the issues brought forth, and I recommend the manuscript for publication.

Reviewer #1 (Remarks to the Author):

The revised manuscript by Yu et al. addressed my comments. The strategy chosen is to drown the reader (me) with figure after figure. This does not improve reading nor does it help the authors to provide a clear and compelling case for their major claim(s). As written in their rebuttal TRIB3 has been described to be involved in any possible mechanism depending on the tumor type (see quote from rebuttal below). My interpretation hereof is that apparently any claim can be made regarding TRIB3 each claim is valid and thus what do we learn ??? As mentioned initially in my first review, the essence of most of what is described here (TRIB3 acts on AKT, AKT regulates FOXO, FOXO is regulated through PTMs and UBI, FOXOs in stemness etc. is already known. so the advance presented here to me is minimal.

minor point: the Ubiquitin assay is still technically invalid. These should be performed using strong denaturing conditions (i.e. ip in Urea) in order to minimise the chance that one looks at UBI associated proteins rather than ubiquitinated proteins.

"Recent findings from our group have indicated that TRIB3 interacts with different proteins to support the tumor stemness in various cancers. In colon cancer, the TRIB3/ β -catenin/TCF4 heterodimer promotes colon cancer tumorigenesis and cancer stemness (Hua F et al., 2019, *Gastroenterology*). In acute promyelocytic leukemia (APL), TRIB3 interacts with oncoprotein PML-RAR α and enhances the stability of PML-RAR α , which promotes the self-renewal of APL-initiating cells (Li K et al., 2017, *Cancer Cell*). Moreover, TRIB3 interacts with autophagy protein p62 to promote cancer development in liver cancer, colon cancer, lung cancer and melanoma (Hua F et al., 2015, *Nat Commun*). These studies suggest that each type of cancer has its own key pluripotency factor and regulatory network to mediate or promote high TRIB3-driven tumorigenesis. If so then any claim on TRIB3 function "

Re: We agree with your criticisms. Following your suggestions, we soft our tone through manuscript and change our main conclusion as providing a link between a stress sensor pseudokinase TRIB3 and breast cancer stemness with the reported key pluripotency factors. Particularly, we have indicated that TRIB3 interacts with AKT1 in liver to regulate insulin resistance (Du K et al., 2003 *Science*, Line 510), the role of FOXO1 in embryonic stem cell as reported by Zhang X et al. (2011 *Nat. Cell Biol*, Line 481), and SOX2 can be upregulated by FOXO1 to support stemness in embryonic cells (2011 *Nat. Cell Biol*, Line 483). Thus, in the first paragraph of Discussion section, we only state that the stress-induced TRIB3 expression activates the AKT1-FOXO1-SOX2 axis by interacting with AKT1 and interfering with FOXO1 degradation in supporting SOX2-driven breast cancer stemness in TRIB3-overexpressed BCCs. Indeed, our study also provides evidence to show that therapeutics targeting this axis by Pep2-Ae, an AKT-derived alpha helix peptide, can reduce breast cancer stemness and progression.

Regarding the Ubiquitin assay experiments, we thank you for your constructive suggestion. Following your opinion, we have conducted Ubiquitin assay by using strong denaturing conditions (6 M Urea) to focus on ubiquitinated proteins. The quality of ubiquitination results is significantly improved (Revised Fig. 5g, 5h, Revised Supplementary Fig. 5e-g, Revised Supplementary Fig. 6e). We removed or replaced some unnecessary ubiquitination results and provided detail in the method section. We tried our best to present a clear story and only mended evidence to address all reviewers' concern. All the constructive comments indeed improved our manuscripts. In order to avoid the redundancy, we revised the manuscript as concise as we can and deleted the unnecessary figures based on reviewers' suggestion.

Reviewer #2 (Remarks to the Author):

The authors have satisfactorily addressed my concerns including the inclusion of new data which significantly improves the work.
Some of the English requires editing.

Re: We thank your constructive comments regarding our manuscript. We carefully checked the whole manuscript for any typo and polish all sections in current manuscript.

Reviewer #3 (Remarks to the Author):

The authors have addressed the issues brought forth, and I recommend the manuscript for publication.

Re: We thank this reviewer for his encouraging comments.

REVIEWERS' COMMENTS:

Reviewer #1 (Remarks to the Author):

No further comments